# Quantifying the time of emergence of the anthropogenic signal in the global land carbon sink

Na Li[1, 2], Sebastian Sippel[2], Nora Linscheid[1, 3], Miguel D. Mahecha[3], Markus Reichstein[1], and Ana Bastos[1, 3]

[1]Max Planck Institute for Biogeochemistry, 07745 Jena, Germany.
[2]Institute for Meteorology, Leipzig University, 04103 Leipzig, Germany.
[3]Institute for Earth System Science and Remote Sensing, Leipzig University, 04103 Leipzig, Germany.
[*]Address correspondence to: na.li@uni-leipzig.de

**Abstract.** The global land carbon sink has increased since the pre-industrial period, driven by the increasing atmospheric $CO_2$ concentration and physical processes influenced by climate change. However, detecting these anthropogenic signals in the global land carbon sink is challenging due to the large year–to–year variability, which can mask or amplify long-term trends, particularly on regional and decadal scales. This study aims to detect the time it takes for long-term trends driven mostly by anthropogenic signal to dominate over natural variations, that is, the "time of emergence", in the land carbon sink.

For this, we use five large ensembles of historical simulations (1851–2014) and future scenarios (2016–2100) from Earth system models (ESMs). Our results show that, firstly, the anthropogenic signal in the global net land carbon sink emerges from 26 to 66 years in the period 1960–2009 (relative to the natural variations in the period of 1930–1959), depending on the ESM considered. The time of emergence is considerably shorter for the two major gross carbon fluxes: 8–13 years for gross primary productivity and 6–10 years for total ecosystem respiration. Furthermore, we find that long-term trends in the net land carbon sink at most regional scales take at least 20 years longer to emerge than at the global scale, due to the larger contributions from internal climate variability at smaller scales.

Secondly, future scenarios show delayed signal detection compared to historical trends. This delay is mainly due to weaker anthropogenic signal trends rather than stronger natural variability. The weaker signal reflects primarily the slow-down of the increasing net land carbon sink in response to emission mitigation.

Thirdly, we apply dynamical adjustment to filter out the year–to–year circulation-induced variability in both the historical and future simulations. This approach substantially shortens the detection time for the global net land carbon sink: between 34–39% for the historical period and 29–55% for the future simulations. This approach can also shorten the detection time for observational based datasets (30% reduction in the period 1960-2009), thereby improving our ability to detect long-term trends of land carbon sink variability. Given that long-term trends are mostly associated with human impacts on the land carbon cycle, our proposed approach can offer valuable insights on the effectiveness of policy decisions and their implementation.

# 1 Introduction

The global land carbon sink has been increasing since the pre-industrial period (Friedlingstein et al., 2022; Ruehr et al., 2023), mainly driven by the increasing atmospheric $CO_2$ and mid- to high-latitude warming caused by human activities (O'Sullivan et al., 2022). Detecting such anthropogenic signals in observations of annual atmospheric $CO_2$ concentration remains challenging due to the large year–to–year natural variations, which can obscure or enhance long-term trends, especially at regional scales and for shorter periods (Deser et al., 2012b; Kay et al., 2015; Maher et al., 2019; Chen et al., 2021; Bonan et al., 2021).

The global net land carbon sink refers to the balance between carbon absorption through gross primary productivity (GPP, photosynthesis at large scale) and carbon release through total ecosystem respiration (TER), but also through fires and other disturbances (Canadell et al., 2021; Ciais et al., 2022). GPP and TER are directly driven by local climate variability, such as temperature and precipitation (Jung et al., 2017; Piao et al., 2020; Canadell et al., 2021). Elevated atmospheric $CO_2$ concentrations have contributed to an increase in the global land carbon sink (Ruehr et al., 2023) through increasing GPP (Walker et al., 2021). Warming temperatures, particularly at high latitudes, have also contributed to increasing GPP (Ruehr et al., 2023).

The long-term trends of the global carbon cycle are superimposed with substantial year–to–year variations (Piao et al., 2020). These variations mostly originate from natural processes, including internal climate variability—fluctuations across a continuum of time scales—as well as from natural external forcings such as volcanic eruptions and solar radiation (Deser et al., 2012b; Canadell et al., 2021; Eyring et al., 2021; Mercado et al., 2009; Zhang et al., 2021). Internal climate variability is often regarded as an irreducible noise within the signal of long-term forced climatic trends, arising from internal atmospheric dynamics and from atmosphere-ocean interactions (Deser et al., 2012a, 2020; Lehner et al., 2017; Bonan et al., 2021). Such variability manifests both as short-term weather events and as low-frequency climate patterns, such as the El Niño/Southern Oscillation (ENSO) which strongly influence global land carbon sink variations through associated changes in temperature and precipitation (Bacastow, 1976; Keeling et al., 1995; IPCC, 2021; Li et al., 2022).

The detection of anthropogenic signals in the global land carbon sink is important for improving our understanding of carbon-climate feedback and refining future carbon projections (Friedlingstein et al., 2014). Detection involves identifying a statistically significant "signal" of long-term forced changes against the "noise" of natural variability in the system (Chen et al., 2021) and is important for improving our understanding of carbon-climate feedback and refining future carbon projections (Friedlingstein et al., 2014). However, several fundamental challenges remain:

First, internal climate variability can be realized differently in multiple simulations under the same external forcings, which may be seen as random and difficult to predict (Frankcombe et al., 2015; Deser et al., 2020; Doblas-Reyes et al., 2021; Bonan et al., 2021). Since observations only represent one unique realization of internal climate variability, they are insufficient to characterize the full range of physically plausible internal climate variability. Moreover, internal climate variability is sensitive to the choice and length of the study period (Kumar et al., 2016; Doblas-Reyes et al., 2021; Maher et al., 2024), making it harder to separate natural fluctuations from forced signals (Bonan et al., 2021; Frankcombe et al., 2015; Doblas-Reyes et al., 2021). This makes it challenging to capture the full dynamics of internal climate variability, particularly due to the limited length of observation records (Maher et al., 2019; Chen et al., 2021).

Second, ecosystem responses vary across geographic regions and timescales of natural climate variations and forcing (Lombardozzi et al., 2014). Regions with high natural climate variability might not show high land carbon sink variability (Lombardozzi et al., 2014). The detection and attribution of anthropogenic signals thus strongly depend on the specific regions of interest (Deser et al., 2012b; Hawkins and Sutton, 2012; Deser et al., 2012a; Mahlstein et al., 2012; Lombardozzi et al., 2014).
On decadal time scales, internal climate variability in land-atmosphere $CO_2$ flux often mask the anthropogenic signals in many regions (Lombardozzi et al., 2014; Kumar et al., 2016; Doblas-Reyes et al., 2021; Bonan et al., 2021).

Large ensembles of Earth system model (ESM) simulations with perturbed initial conditions are effective tools to address these challenges (Deser et al., 2020; Bonan et al., 2021). By running sufficient simulations in a single model with slightly different initial conditions, and under the same physical process representation and external forcing, the distribution of internal climate variability is sampled more effectively than with a single realization (Milinski et al., 2020; Chen et al., 2021). The externally perturbed signal (dominated by anthropogenic signal) emerges as the ensemble mean, that is, a deterministic signal (Milinski et al., 2020; Deser et al., 2020). The residual after removing the ensemble mean can thus be regarded as mostly internal natural variability in the climate system (Milinski et al., 2020; Deser et al., 2020; Bonan et al., 2021). Based on such large ensembles of ESM simulations, the "time of emergence (ToE)" can be determined as the time required for an external perturbed signal (mostly anthropogenic-caused climate change) to become larger than the amplitude of natural variations (Hawkins and Sutton, 2012; Lehner et al., 2017; Schlunegger et al., 2020; Bonan et al., 2021). The ToE metric helps to identify climate change impacts on regional and global scales, and attribute changes to particular causes (Chen et al., 2021). However, due to large year–to–year variations, the anthropogenic signal may remain within the range of natural variability for multiple decades (Lombardozzi et al., 2014; Bonan et al., 2021; Ranasinghe et al., 2021).

Here, we evaluate how long it takes for long-term trends in the global land carbon sink—primarily driven by anthropogenic perturbations—to be detected at different spatial scales. To achieve this, we estimate the ToE in ESM simulations under historical and future scenarios. Our key objectives are to: 1) detect the anthropogenic perturbed signal in global land carbon sink in historical simulations (1851-2014); 2) examine the spatial effects in the ToE on regional scales; 3) estimate the ToE under various future scenarios (2016-2100) and 4) test approaches to separate circulation-induced variability in the ToE in the global land carbon sink.

## 2 Methods and dataset

In this study, we use five ESM large ensembles to investigate the time to detect anthropogenic perturbed signals in global and regional land carbon sinks.

### 2.1 Dataset

We use outputs from historical simulations by ESMs with at least 30 realizations to investigate the ToE in the land carbon sink. The models selected include the CESM2-LE with 90 simulations (Danabasoglu et al., 2020; Rodgers et al., 2021) and four models in CMIP6 (Eyring et al., 2016; Brunner et al., 2020): ACCESS-ESM1-5 with 38 simulations (Ziehn et al., 2020),

CanESM5 with 40 simulations (Swart et al., 2019), IPSL-CM6A-LR with 33 simulations (Boucher et al., 2020), and MPI-ESM1-2-LR with 41 simulations (Mauritsen et al., 2019). All historical simulations are conducted under the CMIP6 historical forcing, including volcanic eruptions and changes in atmospheric composition due to human activities (Eyring et al., 2016). The future scenario simulations are modeled under different Shared Socioeconomic Pathways (SSPs), based on varying levels of human-emitted $CO_2$ and mitigation efforts (Chen et al., 2021; Lee et al., 2021; O'Neill et al., 2016).

The historical simulations covers the period of 1851-2014 and the future scenario simulations cover the period from 2015 to 2100. The spatial resolution of CESM2-LE outputs is $0.9375° \times 1.25°$, and four CMIP6 models is $2.5° \times 2.5°$ (pre-processed by Brunner et al. (2020) from their native spatial resolution). We select the net biome production (NBP), gross primary production (GPP), and total ecosystem respiration (TER) from the above five ESMs. Note that the TER in CESM2-LE is calculated according to Eq. 1, where TER is estimated as the difference between GPP, primary production (NPP), corresponding to autotrophic respiration, soil respiration (SR), and litter respiration (LR) (Eq. 1).

$$TER = GPP - NPP + SR + LR \tag{1}$$

The TER in four CMIP6 models is calculated based on the sum of autotrophic (ra) and heterotrophic respiration (rh) (Eq. 2).

$$TER = ra + rh \tag{2}$$

CESM2-LE outputs of NBP, GPP, NPP, SR, and LR are downloaded from https://www.earthsystemgrid.org/dataset/ucar.cgd.cesm2le.lnd.proc.monthly_ave.html, last accessed on July 11, 2024. For the other four CMIP6 models, NBP, GPP, ra and rh are downloaded (originally from https://esgf-node.llnl.gov/projects/cmip6/) then pretreated by Brunner et al. (2020), last accessed on July 11, 2024. We further download the monthly mean sea level pressure (SLP) from the five models from their respective sources.

For the regional analysis, we use the regional carbon cycle assessment and processes (RECCAP-2) (Ciais et al., 2022) map (https://www.bgc-jena.mpg.de/geodb/projects/Data.php) that categorizes the global land surface into 10 distinct domains, with resolution of $0.5° \times 0.5°$.

We also included the observations of atmospheric $CO_2$ growth rate (AGR) at Mauna Loa (Lan et al., 2025) from 1960 to 2009, downloaded from https://gml.noaa.gov/webdata/ccgg/trends/co2/co2_gr_gl.txt, last accessed on August 18th, 2025. We used monthly mean SLP from the ERA5 reanalysis dataset (Hersbach et al., 2023) for the period 1959–2009, with resolution of $0.25° \times 0.25°$, downloaded from https://cds.climate.copernicus.eu/datasets/reanalysis-era5-single-levels-monthly-means?tab=overview, last accessed on August 18th, 2025.

## 2.2 Data pretreatment

NBP, GPP, and TER from CESM2-LE are provided in the unit of $gC \cdot m^{-2} \cdot s^{-1}$, from which an annual sum is calculated. NBP, GPP, and TER from four CMIP6 models are in unit of $kgC \cdot m^{-2} \cdot s^{-1}$ and converted to annual sums in $gC \cdot m^{-2} \cdot yr^{-1}$. TER is

calculated according to Eq. 1 for CESM2-LE and according to Eq. 2 in the four CMIP6 models. In order to have consistent sign with GPP, TER here is multiplied by –1. In the historical simulations (1851–2014), NBP, GPP, and TER of the five model datasets are area–weighted and aggregated to domain mean with the spatial resolutions of $2.5°\times2.5°$, $5°\times5°$, $10°\times10°$, $20°\times20°$, $30°\times30°$, $45°\times45°$, $60°\times60°$, and global mean. The global mean of NBP, GPP, and TER is also calculated for the four future scenarios, with period of 2016–2100 selected (in CMIP6 models the time series starts at July 2015, so we select from 2016 instead). Note that CESM2-LE only includes one future scenario (SSP3-7.0), and other models included all four future scenarios. SLP from all five ESMs is aggregated to the resolution of $10°\times10°$. Data pre-processing, including unit conversion and spatial aggregation, was performed with the Climate Data Operators software (Schulzweida, 2023, CDO).

The RECCAP-2 map is area–weighted and aggregated to $2.5°\times2.5°$, then categorize the NBP, GPP, and TER to 10 RECCAP-2 regions.

The pretreatment steps of atmospheric $CO_2$ growth rate (AGR) at Mauna Loa from 1960 to 2009 (Lan et al., 2025) follows Li et al. (2022). We first remove five volcanic years (1963, 1982, 1983, 1991, and 1992), then fitted the long-term trend with locally weighted scatterplot smoothing (Cleveland et al., 1991, LOWESS). SLP from ERA5 (Hersbach et al., 2023) also have five volcanic years removed, then area–weighted and aggregated to the spatial resolution of $9°\times9°$.

## 2.3 Methods

### 2.3.1 Time of emergence

To determine the time of emergence (ToE), we apply the noise–to–signal ratio approach, following Bonan et al. (2021). The signal (S) refers to the anthropogenic perturbation driven response, which is the linear regression slope of the ensemble mean of the simulations for each model (Bonan et al., 2021). For the calculation of N, we first select all years across all model simulations over the selected period, then mix the data from all years in the selected period together and calculate the standard deviation. In the historical simulations, the noise (N) corresponds to the standard deviation of the ensemble before the 1960s (here is 1930–1959), a period less affected by human activities compared to more recent ones, and used as the baseline for natural variability (Bonan et al., 2021). In the future scenarios, we calculate the ToE for NBP, GPP, and TER, with the signal period in 2020–2070 and the noise period in 2020–2070 (with the ensemble mean removed). ToE (Eq. 3) represents the time needed for the anthropogenic perturbed signal to become larger than the amplitude of the noise (Bonan et al., 2021).

$$ToE\ (years) = 2N/S \tag{3}$$

Here we use a linear regression slope rather than a nonlinear approach to represent the signal trend, this is to capture the dominant forced signal in the selected signal period. The ensemble mean of NBP, GPP and TER reflects the forced ecosystem response, including anthropogenic forcing, short-period natural forcings (e.g., volcanic eruptions), and decadal internal variability (Deser et al., 2012b; Canadell et al., 2021; Eyring et al., 2021; Mercado et al., 2009; Zhang et al., 2021). The linear trend captures the first-order (Hasselmann, 1979) long-term anthropogenic influence, whereas nonlinear methods could risk over-

fitting and mis-attributing natural forcing or internal variability to anthropogenic signals, especially at regional scales where variability is larger (see Fig. 3 and appendix A Fig. A.1).

### 2.3.2 Noise filtering based on dynamical adjustment

To shorten the detection time, we use a dynamical adjustment technique to estimate circulation-induced variability in NBP. Dynamical adjustment is a technique in climate science, which aims to isolate circulation-induced variability (such as in temperature and precipitation); where the residual time series in those climate variables is thought to contain the forced response (Smoliak et al., 2015; Deser et al., 2016; Sippel et al., 2019). Circulation-induced variability is generally expected to reflect internal climate variability to the largest extent (Deser et al., 2016; Smoliak et al., 2015; Sippel et al., 2019). Therefore, dynamical adjustment allows one to obtain a higher signal–to–noise ratio in the circulation-filtered residual time series, where the residual represents the remainder after subtracting the estimated circulation-induced variability from the target variable.

Here, we employ ridge regression, a dynamical adjustment technique, to estimate circulation-induced variability (Sippel et al., 2019). In our model, the sea level pressure (SLP) field is used as a predictor and proxy of circulation-induced variability (Sippel et al., 2019). As a regularized linear regression method, ridge regression allows for including full spatiotemporal dynamics of circulation variations while overcoming multicollinearity and overfitting, which typically arise from a large number of predictors and relatively short study period (Hastie et al., 2009; Sippel et al., 2019). This approach was adapted by Li et al. (2022) to evaluate the fraction of atmospheric circulation-induced variations in global carbon cycle variability. The key steps include (Sippel et al., 2019; Li et al., 2022): 1) Select pixel based time series of global SLP, to be used later for predicting global carbon cycle variability. We then calculate the mean seasonal SLP. Because DJF (December–February) SLP provides the highest predictability of annual NBP (see Li et al. (2022) for details), we use DJF SLP in this study. 2) Select the time series representing global land carbon variability; here, this corresponds to the global annual NBP with the ensemble mean removed. 3) Training and testing. Here, the first half of the dataset is used for training and the second half for testing. For example in historical simulations, the training data is the time series from 1851 to 1932, and the testing data is in 1933–2014. 4) Switch the training and testing data to start a new round of model training and prediction. This means, the training data from step 3 is used as testing data, and the testing data from step 3 is used as training data. Then we have the full time series of NBP that is predicted by DJF SLP. Detailed model design can be found in Sippel et al. (2019); Li et al. (2022).

By using DJF SLP to predict NBP (with the ensemble mean removed), we estimate the fraction of circulation-induced variability in global NBP time series. The residual, after removing the DJF SLP predicted NBP, reflects mostly the influence of natural forcing (e.g., volcanic eruptions or solar radiation variability), disturbances (fires, when simulated by models), and unpredictable high frequency internal climate variability (Sippel et al., 2019; Piao et al., 2020; Canadell et al., 2021). We hypothesize that this method reduces noise levels in NBP and allows for an earlier detection of the anthropogenic signal.

### 2.4 Statistical analysis

We perform four statistical analyses: 1) ToE in land carbon fluxes from historical simulations. We analyse the ToE of the anthropogenic perturbed signal in NBP, GPP, and TER in the historical simulations. Following Bonan et al. (2021), the signal

(S) is the linear regression slope of the ensemble mean in the period of 1960–2009, and noise (N) is the standard deviation of all simulations in the period 1930–1959. We first compare the historical time series of NBP, GPP, and TER, and then calculate the ToE according to Eq. 3. 2) Spatial effects on ToE. We examine how the ToE varies globally and across the 10 RECCAP-2 regions. In addition, we evaluate the influence of spatial resolution on ToE. We calculate pixel-based ToE values at multiple spatial scales (ranging from $5°\times5°$ to $60°\times60°$) and compare these with the global scale. 3) ToE in future projections of the land carbon fluxes. We calculate the ToE for NBP, GPP, and TER, with the signal period in 2020–2070 and the noise period in 2020–2070 (with the ensemble mean removed). 4) Noise reduction through dynamical adjustment. Given the large year–to–year variability in NBP, we use ridge regression to remove the circulation-induced variability in global NBP. To assess the effectiveness of ToE reduction on a global scale through dynamical adjustment, we calculate the relative reduction ($d_S$) according to Eq. 4.

Note that only the calculated signal (regression slope) with significance value $P < 0.05$ is selected. If the calculated signal (regression slope) is negative, we then take the absolute signal value to get a positive ToE. Here we select to show the ToEs less than 150 years.

$$d_S = 100\% * (V_O - V_R)/V_O \tag{4}$$

Note that $V_O$ represents the original value (ToE or N) and $V_R$ is the (ToE or N) estimated from the original time series (NBP or GPP) after removing the circulation-induced variability estimated by using the ridge regression model.

We also calculated the contribution of N and S to ToE changes in each RECCAP-2 region, relative to the global scale, for NBP, GPP, and TER. Eq. 5 is the natural logarithmic form of equation Eq. 3. We first calculate the logarithmic changes on the global and regional scales, and then calculate the differences between each region and the global mean (Eq. 6).

$$ln(ToE) = ln(2 \times N) + ln(1/S) \tag{5}$$

$$ln(ToE_{region}) - ln(ToE_{global}) = ln(2 \times N_{region}) - ln(2 \times N_{global}) + ln(1/S_{region}) - ln(1/S_{global}) \tag{6}$$

The contribution of changes in N and S are:

$$N_{contri} = 100\% \times \frac{ln(2 \times N_{region}) - ln(2 \times N_{global})}{ln(ToE_{region}) - ln(ToE_{global})} \tag{7}$$

$$S_{contri} = 100\% \times \frac{ln(1/S_{region}) - ln(1/S_{global})}{ln(ToE_{region}) - ln(ToE_{global})} \tag{8}$$

Note that for future scenarios, we substitute the region's value to each future scenario's value.

## 3 Results and discussion

### 3.1 Detection of anthropogenic signal in historical simulations

We first examine the NBP time series for the historical simulations from 1851 to 2014 (Fig. 1). Before the 1960s, the ensemble mean (long-term trend) for each model remains relatively stable with slight variations. After the 1960s, the ensemble mean shows a noticeable increase. Despite this, the magnitude of NBP variability remains consistent or slightly increase throughout the historical period, for all models. In individual simulations, we observe that the year–to–year variations are considerably larger than the changes in the ensemble mean, enhancing or offsetting the long-term NBP trend (Fig. 1).

We then examine the time series of GPP and TER in the historical simulations (Fig. 1). Both GPP and TER show similar trends across models, though ACCESS-ESM1-5 shows a larger magnitude difference (Fig. 1). The ensemble mean of GPP and TER are similar until the 1960s, after which GPP slightly surpassed TER (Fig. 1). Year–to–year variations are minor compared to the long-term trend in the ensemble mean, suggesting that photosynthesis and respiration are strongly influenced by anthropogenic perturbations. Because the trends in GPP and TER largely compensate when combined to calculate NBP, the resulting NBP exhibits smaller long-term trends but pronounced interannual variability.

### 3.2 Spatial effects of ToE

We then examine how long it takes for the anthropogenic signal (ensemble mean of each model) to emerge from year–to–year variations of NBP in global scale and across 10 RECCAP-2 regions (Fig. 2b). Globally, CESM2-LE has the shortest detection time at 26 years, while CanESM5 takes the longest at 66 years (Fig. 2b). The detection time in ACCESS-ESM1-5 is not available, due to a flat trend of ensemble mean after 1960s.

We first check the ensemble mean of NBP in global scale and 10 RECCAP-2 regions (Appendix A Fig. A.1). The trends of ensemble mean in NBP subjects to larger interannual variability in regional than in global scales, particularly in Southeast Asia and Africa. Comparing noise (N) and signal (S) (Appendix A Fig. A.4 and Eq. 3), we found that the variation in detection time across models is mainly due to differences in year–to–year variability and signal trends.

In most of the 10 RECCAP-2 domains, ToE detection in NBP takes longer than at the global scale (Fig. 2b). We calculate the contribution of changes in N and S to regional ToE, compared with historical global scale NBP (Appendix A Fig. A.5). In regions such as South Asia and Australasia, the longer regional ToE is mainly due to larger regional noise (Appendix A Fig. A.5). While in other regions such as East Asia, the longer regional ToE is mainly attributable to smaller regional S (Appendix A Fig. A.5). These effects delay the detection of anthropogenic signals, a phenomenon we refer to as "spatial delay".

The spatial delay in NBP can be explained by the reduced noise from internal climate variability when fluxes are aggregated globally, while the signal trend may be either enhanced or diminished depending on the specific region considered. (Appendix A Fig. A.4, A.5). This is a well-known pattern in detection and emergence studies in the climate literature (Mahlstein et al., 2011; Lehner et al., 2017). However, the spatial delay does not apply everywhere. In Russia, models like CanESM5, and

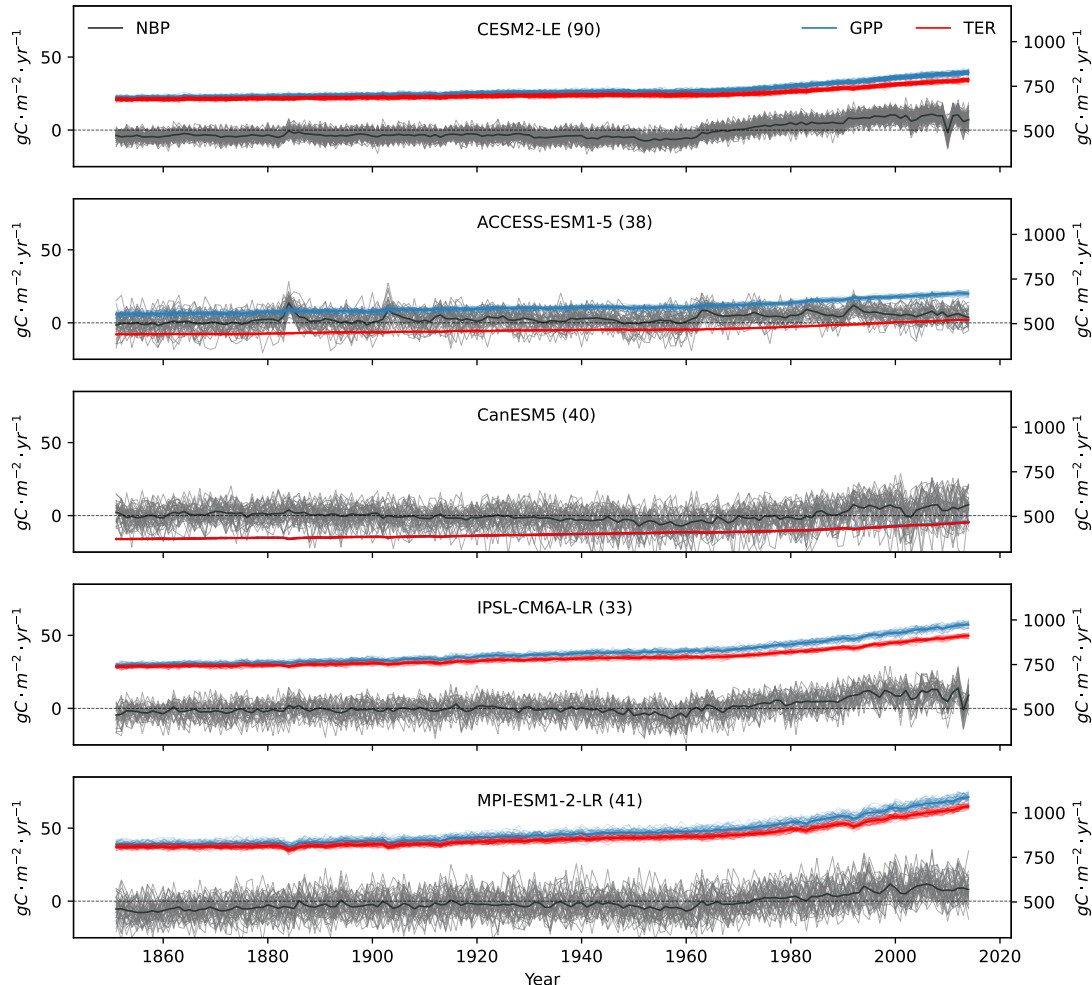

**Figure 1.** Time series of NBP, GPP, and TER from 1851 to 2014 in five ESM large ensembles. The thin lines represent individual simulations, while the bold lines represent the ensemble mean. The gray lines show NBP, corresponding to the left y-axis. The blue and red lines corresponds to the right y-axis and represent GPP and TER, respectively. The number of simulations for each model is listed in the legend next to the model name. Note that TER in model ACCESS-ESM1-5 only included 24 simulations, due to limited data availability.

IPSL-CM6A-LR show shorter detection time compared to the global scale (Appendix A Fig. A.4, A.5). This is mainly due to
apparent smaller noise in Russia of the two models, compared with global scale and other regions (Appendix A Fig. A.4, A.5).

We then evaluate the ToE for GPP and TER (Fig. 2c, d), both show similar patterns in detection time, and the relative importance of noise and signal at regional scale (Appendix A Fig. A.6, A.7, A.8, A.9). Globally, it takes around 8 to 13 years to detect anthropogenic signal in GPP and 6 to 10 years for TER (Fig. 2c, d). As found for NBP, both GPP and TER show a spatial delay from global scale to regional scale (Fig. 2c, d). For GPP and TER, longer regional ToEs are mostly due to larger regional noise rather than weaker signal trends, except in West Asia, where it is mainly driven by an apparent weaker signal trend (Appendix A Fig. A.7, A.9). Australasia in GPP and TER both generally have the longest detection time, due to higher noise compared to lower signal levels (Appendix A Fig. A.6, A.8), indicating higher internal climate variability. In both GPP and TER, South America and Southeast Asia experience high levels of noise and signal, while West Asia has relatively low levels of noise and signal (Appendix A Fig. A.6, A.8).

Compared with the land carbon sink (NBP), photosynthesis (GPP) and ecosystem respiration (TER) individually show a much shorter detection time of the anthropogenic signal (Fig. 2). This is likely due to the fact that GPP and TER trends are strongly influenced by anthropogenic perturbations, with the magnitude of the trend exceeding the magnitude of internal climate variability in a much shorter time. However, when calculating NBP, the long-term trends of GPP and TER offset each other, leaving NBP with weaker long-term trends relative to the year–to–year natural variations, thus making it harder to detect the anthropogenic signal in NBP.

To further analyse the spatial delay effect, we calculate the distribution of pixel based ToE for NBP, GPP, and TER under varying resolutions in the historical simulations. For NBP, as the resolution becomes coarser, the spread of the ToE distribution decreases substantially (Fig. 3), though the median remains similar. This might be due to noise reduction by spatial aggregation through offsetting internal climate variability (Lombardozzi et al., 2014) (Fig. 3). A similar pattern is observed in GPP and TER, where aggregation reduces the spreads of ToE without substantially altering the medians (Appendix A Fig. A.10 and A.11).

We found global scale takes shorter time to detect long-term trends induced by anthropogenic effects than at regional scales, with ToEs increasing for smaller domains as reported by Lombardozzi et al. (2014), though their study used fewer models and less than 10 simulations. A few regions, however, show shorter ToEs than the global scale. For example, in Russia, CanESM5 and IPSL-CM6A-LR simulate relatively small noise and stronger signal trends, leading to shorter ToEs. This maybe relate to the sparsely distributed ecosystems included in models, which are less sensitive to changes in climate drivers. We found that, for regional NBP, larger interannual variability in the signal trend also contributes to longer detection time, likely reflecting different regional climate drivers (e.g., fires, decadal internal variability, land use changes (Deser et al., 2012b; Canadell et al., 2021; Eyring et al., 2021; Mercado et al., 2009). Such large signal variability in regions like Southeast Asia and Africa therefore introduces substantial uncertainties in detecting the anthropogenic signal on decadal timescales.

The large interannual variations in NBP largely arise from variations in GPP and respiration. As regional ecosystems are more sensitive to precipitation than to temperature (Jung et al., 2017), much of this variability maybe influenced by precipitation (Humphrey et al., 2018, 2021). However, anthropogenic signals in precipitation are less robust and emerged later than those in temperature (Doblas-Reyes et al., 2021). Identifying these signals and their impacts on regional ecosystem activity could therefore enable a cleaner and earlier detection of anthropogenic influences on land carbon sinks.

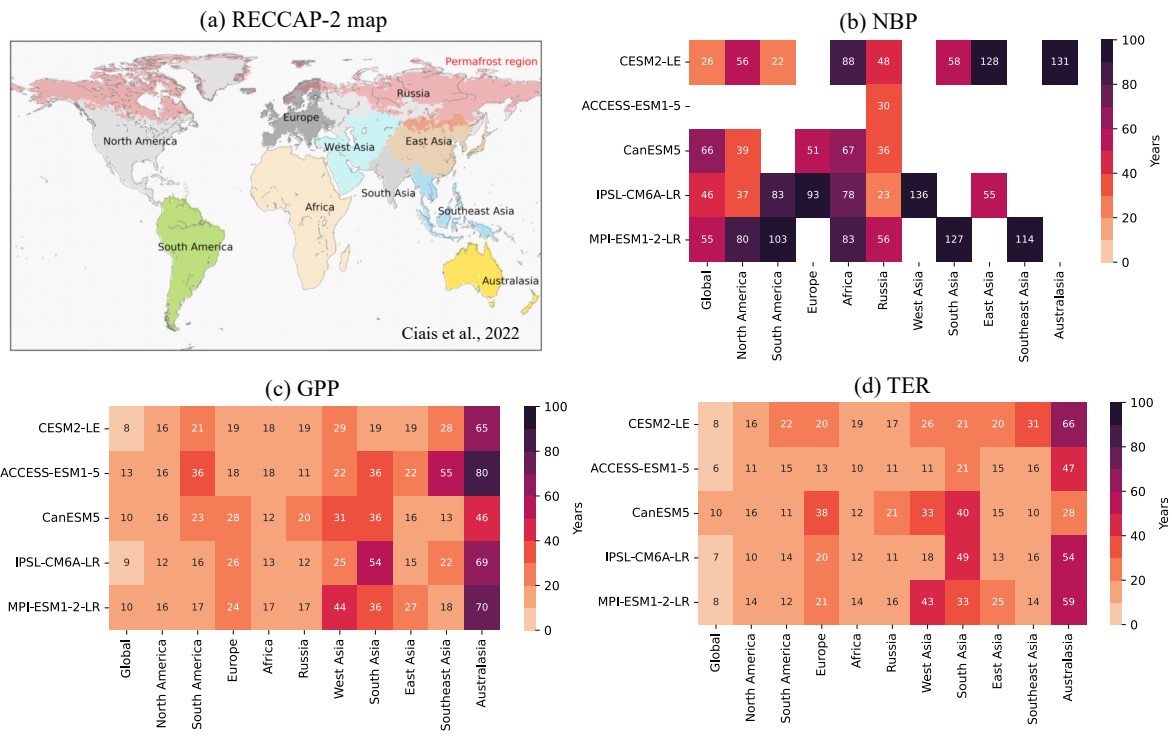

**Figure 2.** ToE of NBP on a global scale and across 10 RECCAP-2 regions, under historical simulations of five ESM large ensembles. Note that ToE is the years detectable after 1960, and is calculated with signal period of 1960-2009 relative to the noise period of 1930-1959, details please check Sect. 2.4. **(a)** RECCAP-2 map (duplicated from Ciais et al. (2022) Fig. 1) that divides the global continents into 10 domains. Note that the RECCAP-2 map is aggregated from $0.5^\circ \times 0.5^\circ$ to $2.5^\circ \times 2.5^\circ$, the spatial domains are slightly changed **(b)** Heat map of the ToE in global and each spatial domain of NBP. **(c)** and **(d)** are heat maps of the ToE in global and each spatial domain of GPP and TER separately. Domains with no significant signal ($P > 0.05$) or ToE longer than 150 years are shown as empty squares.

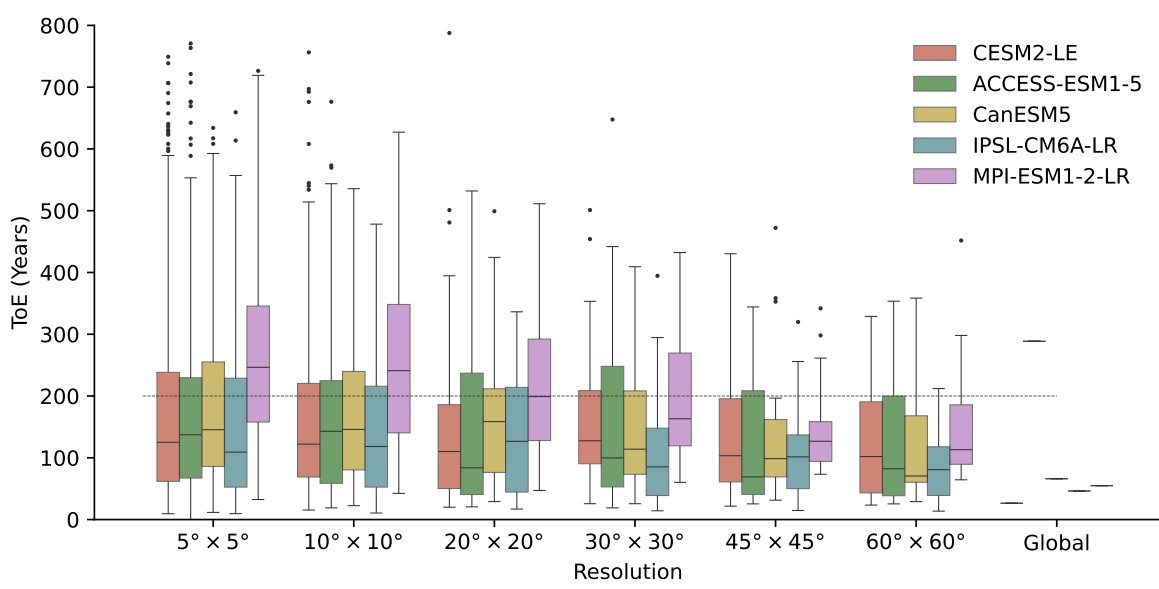

**Figure 3.** Spatial effect in NBP historical simulations across five ESM large ensembles. The distribution of ToE (years after 1960) is shown for varying spatial resolutions. We aggregate the global data per pixel to different resolutions, then calculate the ToE per pixel. The line within each box indicates the median. Note that all signals are in absolute values, so the calculated ToEs are all positive.

### 3.3 ToE in future projections

We examine the time series of NBP under various future scenarios from 2016 to 2100 (Fig. 4). NBP trends of ensemble mean show large deviations across models (Fig. 4).

In CESM2-LE, the SSP3-7.0 scenario shows a steady increase in NBP until around 2040, followed by stable trend until 2100 (Fig. 4). ACCESS-ESM1-5 and IPSL-CM6A-LR exhibit mixed NBP trends in all scenarios, with a relatively stable trend before 2050 and a gradual decline afterwards—ACCESS-ESM1-5 even shifted to a net carbon source (Fig. 4). In CanESM5, all scenarios are mixed and together increasing until 2050 (Fig. 4). After that, all scenarios diverged according to different emission scenarios (Fig. 4). MPI-ESM1-2-LR also have all scenarios mixed before around 2050, then diverge clearly with a lower overall trend (Fig. 4). Except CanESM5, the large year–to–year variability in NBP makes it challenging to distinguish long-term trends across scenarios.

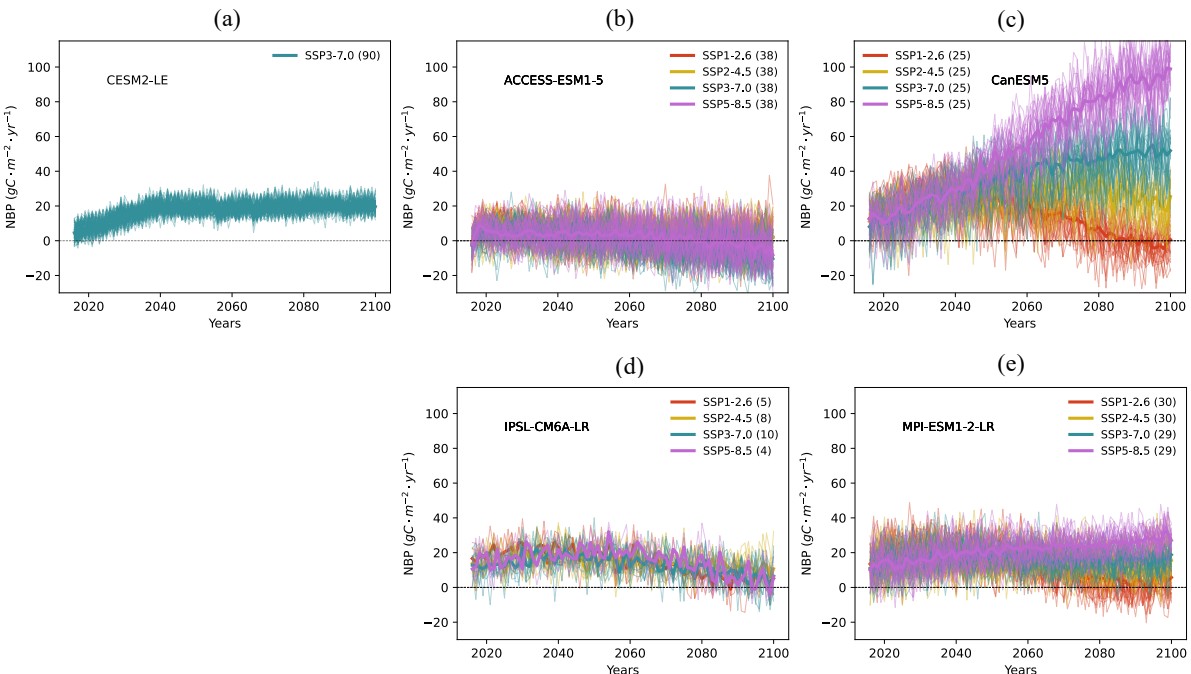

**Figure 4.** The time series of future NBP from 2016 to 2100 across five ESM large ensembles. The four future scenarios include SSP1-2.6 (red line), SSP2-4.5 (yellow line), SSP3-7.0 (green line), and SSP5-8.5 (purple line). Thin lines represent individual simulations, while thick lines represent the ensemble mean for each scenario. The number of simulations for each model scenario is indicated in the legend next to the scenario label.

We then examine the time series of GPP and TER under future scenarios (Appendix A Fig. A.12 and A.13). GPP continues to rise in all models until 2100, except for SSP1-2.6, in which GPP slightly decreases after ca. 2060 (Appendix A Fig. A.12). TER follows a similar pattern, with an increasing trend in line with the different $CO_2$ emission scenarios (Appendix A Fig. A.13). The increase in GPP is likely due to the enhanced $CO_2$ fertilization and warming in mid-to-high latitudes (Ruehr et al., 2023; O'Sullivan et al., 2022).

Distinguishing the ToE for NBP from different future scenarios is challenging, due to smaller anthropogenic signal and larger year–to–year variations across four future scenarios (Fig. 5). Only CanESM5 shows a clear separation between scenarios, with ToE of 147 years for SSP1-2.6, 60 years for SSP2-4.5, 35 years for SSP3-7.0, and 19 years for SSP5-8.5. Other models take over 44 years to detect the anthropogenic signal among all scenarios (Fig. 5). In contrast, GPP and TER trends are more distinct and separated according to different scenarios, resulting in much shorter ToE (Appendix A Fig. A.15, A.17). This might be due to an increase in the $CO_2$ emission level, or a stronger anthropogenic signal that outweighs the increased noise level, making the detection time more driven by impacts from anthropogenic perturbations rather than internal climate variability (Appendix A Fig. A.16, A.18).

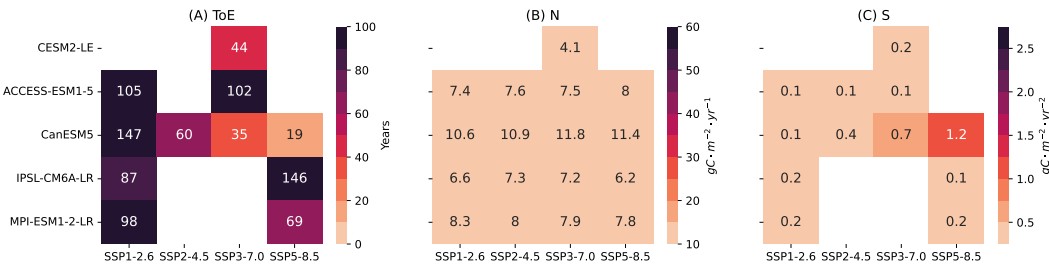

**Figure 5.** Heat map of ToE, noise (N), and signal (S) of global mean NBP under future scenarios.

The divergence of NBP across models is much larger in future scenarios than in historical simulations. While models are inherently different, these differences in the historical period may be amplified in future scenarios due to: (a) the increasing influence of $CO_2$ fertilization and land use change (van Vuuren et al., 2011; O'Neill et al., 2016; Christensen et al., 2018; Arora et al., 2020; Lee et al., 2021; Ciais et al., 2013); and (b) Rising temperature and more frequent extreme events (Friedlingstein et al., 2014; Fischer and Knutti, 2015; Hewitt et al., 2016; Kharin et al., 2018; Vogel et al., 2020; Li et al., 2021; Seneviratne et al., 2021). Both may intensify differences in climate-carbon feedbacks among models (Friedlingstein et al., 2014; Hewitt et al., 2016; Seneviratne et al., 2021).

In future scenarios, it takes longer to detect the anthropogenic signal in NBP, when compared to historical simulations. This delay is mainly due to the small anthropogenic signal caused by the compensation effect of GPP and TER, whose differences are smaller than those in historical simulations (Fig. 5 and Appendix A Fig. A.14-A.18). This may result from a slowdown in the long-term GPP trend under warmer climate and increasing $CO_2$ concentrations in future scenarios. In addition, larger noise levels (Appendix A Fig. A.14) that might driven by more frequent extreme events in a warming climate (Arias et al., 2021) amplifies year–to–year variations in the land carbon sink. Reducing these year–to–year variations is crucial for reducing the ToE in NBP. In the next section, we apply dynamical adjustment to filter out the atmospheric circulation-induced variability in global NBP time series, and assess whether it can contribute to reduce ToE in both historical simulations and future scenarios.

### 3.4 Dynamical adjustment for noise reduction

We use the ridge regression to filter out atmospheric circulation-induced variability in year–to–year global NBP variability (Fig. 6). By applying the ridge regression model that based on sea level pressure as covariates, the circulation-induced variability in the respective carbon flux is predicted. The predicted circulation-induced variability is assumed to contain direct influences (via thermodynamics or $CO_2$ fertilization) of climate change (Sippel et al., 2019). Because circulation-induced variability is highly variable and often assumed to be largely internal variability, the residual can be expected to show a higher signal-to-noise ratio (e.g., Deser et al. (2016); Sippel et al. (2019)).

The noise of global NBP is substantially reduced after filtering out circulation-induced variability, so that ToE is reduced in both historical simulations and future scenarios (Fig. 6, Appendix A Table. A.1). In the historical simulations, the relative

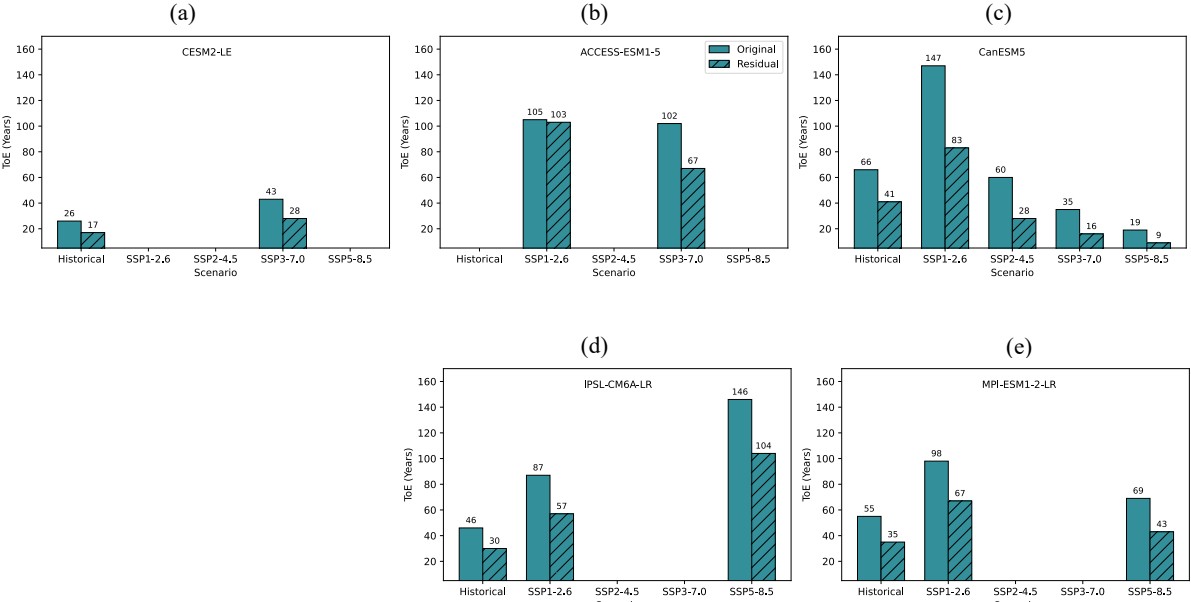

**Figure 6.** ToE of NBP from historical simulations to future scenarios. Note that ToE in historical simulations is calculated with signal period of 1960–2009 relative to the noise period of 1930–1959, and ToE in future scenarios is calculated with signal period of 2020–2070 relative to the noise period of 2020–2070, details please check Sect. 2.4. The solid boxes represent the ToE of NBP, while the hatched boxes represent the ToE of the NBP residual with the circulation-induced variability removed. In cases where both boxes are missing, the respective signal is not available (no significance of linear trend slope), or the ToEs are longer than 150 years.

reduction in ToE ranges from 34% (CESM2-LE) to 39% (CanESM5), corresponding to 9 and 26 years, respectively (Fig. 6, Appendix A Table. A.1). For future scenarios, the reduction ranges from 29% to 55% (42 and 19 years reduction, respectively), except for ACCESS-ESM1-5, where reductions are mostly less pronounced (Appendix A Table A.1). For GPP, the relative reduction in ToE is smaller (Appendix A Fig. A.19 and Table. A.3). In the historical simulations, it ranges from 13% (CanESM5) to 32% (ACCESS-ESM1-5), corresponding to 1 and 4 years, respectively (Appendix A Fig. A.19 and Table. A.3). For future scenarios, the relative reduction ranges from 19% to 60% (1 and 67 years reduction, respectively) (Appendix A Fig. A.19 and Table. A.3). The large reduction of ToE indicates that NBP and GPP are both substantially affected by circulation-induced variability.

We then test the observations of Atmospheric $CO_2$ growth rate (AGR) from Mauna Loa (Lan et al., 2025) for the period 1960–2009, matching the period of signal for the ESM analysis (Fig. 7). The ToE of the observed AGR is 33 years, with noise of 0.87 $gC \cdot yr^{-1}$ and signal of 0.05 $gC \cdot yr^{-2}$ (Fig. 7). After removing circulation-induced variations through dynamical adjustment, the ToE of the adjusted new AGR is reduced to 23 years, with noise of 0.70 $gC \cdot yr^{-1}$ and signal of 0.06 $gC \cdot yr^{-2}$ (Fig. 7). This represents an overall reduction of about 30%, contributed by 19% reduction in noise and 20% increase in signal.

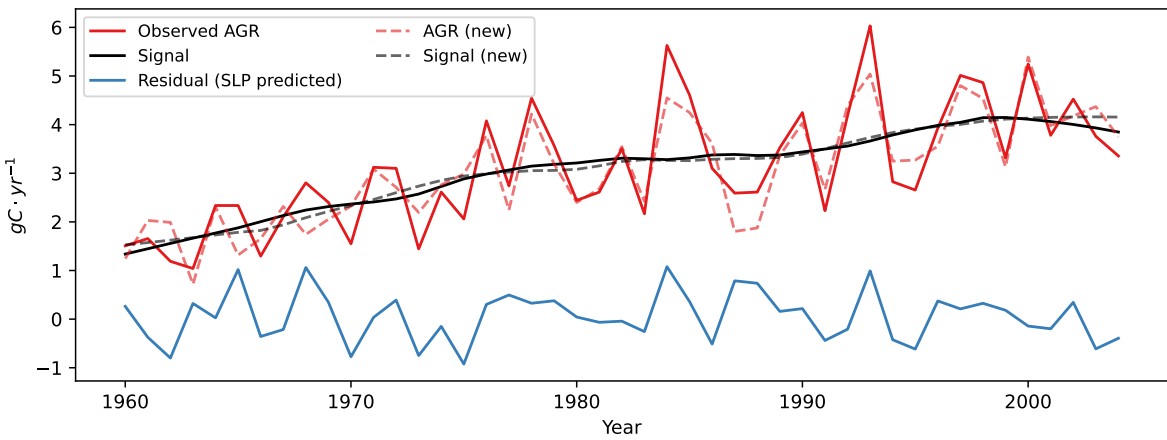

**Figure 7.** Time series of the atmospheric $CO_2$ growth rate (AGR) at Mauna Loa from 1960 to 2009 (Lan et al., 2025). Five volcanic years (1963, 1982, 1983, 1991, and 1992) are removed. The red line is the observed AGR. The black line is the long-term trend fitted with a locally weighted scatterplot smoothing (Cleveland et al., 1991, LOWESS) (signal). The residual (AGR − fitted long-term trend) was predicted using SLP through ridge regression with leave-one-out cross validation (blue line). This SLP predicted residual is then subtracted from the observed AGR to obtain a new AGR time series with circulation-induced variations removed (observed AGR − SLP predicted residual). The dashed black line is the new long-term trend. Data pretreatment and the ridge regression model follow paper Li et al. (2022). Note that the signal period is the same as in models (1960–2009). Due to limited records of $CO_2$ observations before 1958, here we calculate the noise also in the period 1960–2009.

The results show that this approach can be applied in observations, enabling earlier detection of anthropogenic signals in global carbon cycle variability.

## 4  Conclusions

This study examines the detection of long-term trends driven by anthropogenic signals in the global land carbon sink. Using
five ESM large ensembles, we analyze both the historical period (1851–2014) and future scenarios (2016–2100).

In the historical period, the global land carbon sink (NBP) shows large year–to–year variations, which can enhance or obscure long-term anthropogenic trends. While both carbon uptake (GPP) and ecosystem respiration (TER) show trends influenced by anthropogenic perturbations, their year–to–year variations are relatively small. Since NBP corresponds to the balance between carbon absorption (photosynthesis) and release (ecosystem respiration), as well as other fluxes such as fires, the long-term trend
of NBP is in most cases smaller due to this compensation, leaving NBP with a smaller long-term trend and relatively larger year–to–year variations.

We find that the ToE is smaller at global scale compared to regional scales, that is, the anthropogenic signal can be detected earlier at global scale. In the period of 1960-2009, it takes over 26 years for NBP signals to emerge from internal variability,

and around 10 years for GPP and TER. At the regional scale, ToE is longer, which might be due to larger noise from natural climate variability in most regions, as well as detected weaker signal trends. Coarser resolutions reduce the detection time, but the spatial delay is not universal—some high-latitude regions, for example Russia, is found in two CMIP6 models having a shorter detection time of NBP. This is due partly to a smaller noise compared with other regions and the global scale, and partly due to a high signal relative to the small average carbon flux at present in those northern regions. The smaller noise may be also due to the small average carbon flux, and associated small variability.

In future scenarios, it takes longer to detect the anthropogenic signal in NBP, due to lower anthropogenic signal level caused by the compensation effect of GPP and TER, as well as higher noise levels that may result from more frequent extreme events under a warming climate (Friedlingstein et al., 2014; Fischer and Knutti, 2015; Hewitt et al., 2016; Kharin et al., 2018; Vogel et al., 2020; Li et al., 2021; Seneviratne et al., 2021; Arias et al., 2021). The future trends of global land carbon sink differ significantly across models. While some models have time series separated by emissions after 2050, others remain mixed through 2100. This might be due to the large uncertainty in projections of the global land carbon sink (Friedlingstein et al., 2014; Padrón et al., 2022). For high $CO_2$ emission scenarios of SSP3-7.0 and SSP5-8.5, CanESM5 continues to increase after around 2050, while other models show carbon saturation, which may result from model uncertainties related to climate change and nutrient limitations (Arora et al., 2020). Uncertainty in ToE in future projections is closely linked to uncertainties across the model projections of the land carbon sink in the future. In contrast, GPP and TER increase consistently and are well separated by different $CO_2$ emission scenarios.

NBP exhibits larger year–to–year variability and it is difficult to detect the anthropogenic signal. After removing atmospheric circulation-induced variability from NBP, the time of emergence of the anthropogenic signal is significantly reduced. In the historical simulations, the relative reduction in the ToE ranges from 34 to 39%, while in future scenarios it ranges between 29 to 55%. Future NBP is more influenced by anthropogenic perturbations and natural variations (Arias et al., 2021). However, anthropogenic perturbations remain the dominant factor of GPP trends, which determine the time of emergence under all future scenarios. This approach has been applied in observations and shows an early detection of anthropogenic signal in global carbon cycle variability.

The emergence approach used in this study is sensitive to the choice of the periods for defining noise and signal. Moreover, the fitted linear slope of the ensemble mean may misrepresent the true signal trend, particularly at regional scales, due to large forced variability in the ensemble mean (Lombardozzi et al., 2014; Bonan et al., 2021). A better understanding of regional ecosystem responses to anthropogenic signals, along with improved methods that are less sensitive to large regional variability, may help reduce the detected emergence time.

This study highlights how early the anthropogenic impacts on the global land carbon sink can be detected. By using a dynamical adjustment technique to remove atmospheric circulation-induced variability, the detection time can be largely reduced. However, there are still substantial uncertainties across models, with differing patterns and large year–to–year variations (Friedlingstein et al., 2014; Arora et al., 2020). Our proposed approach to use dynamical adjustment to reduce ToE can contribute to enhance our ability to monitor human impacts on land carbon variability and thus support decision making. This approach is particularly helpful for detecting whether recent regional carbon flux trends are driven by internal climate variabil-

ity or forced by climate change. Internally driven trends might not going to be sustained in the near-future, while trends forced
by climate change are expected to continue.

*Code availability.* The python scripts used for this study is available at Li (2025)

*Data availability.* Please check Section. 2.1 for details.

*Author contributions.* Conceptualization by NLi, AB, SS; methodology by NLi, AB, SS, NLin, MR, MM; investigation by NLi, AB, SS, NLin; visualization by NLi; supervision by AB, SS, Nlin, MM, MR; writing original draft by NLi; review by AB, SS, NLin, MM.

*Competing interests.* The contact author has declared that none of the authors has any competing interests.

*Acknowledgements.* AI tools ChatGPT (version GPT-4o mini) and Grammarly are used in this manuscript. They helped with writing, including grammar correction and refining sentences and paragraphs. However, the original scientific ideas are from authors.

**Appendix A**

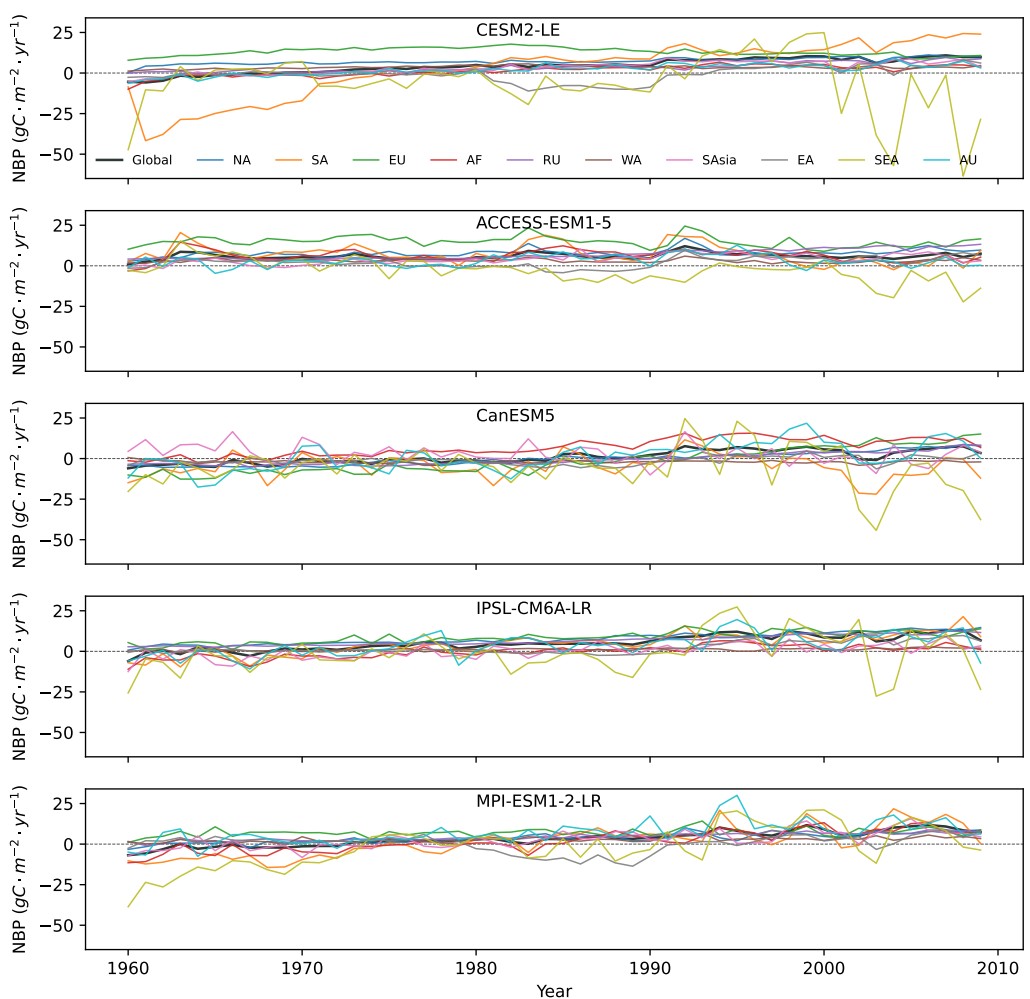

**Figure A.1.** Time series of NBP ensemble mean from five ESMs. The thick black line is the global ensemble mean, and the colored lines represent ensemble means for the 10 RECCAP-2 regions.

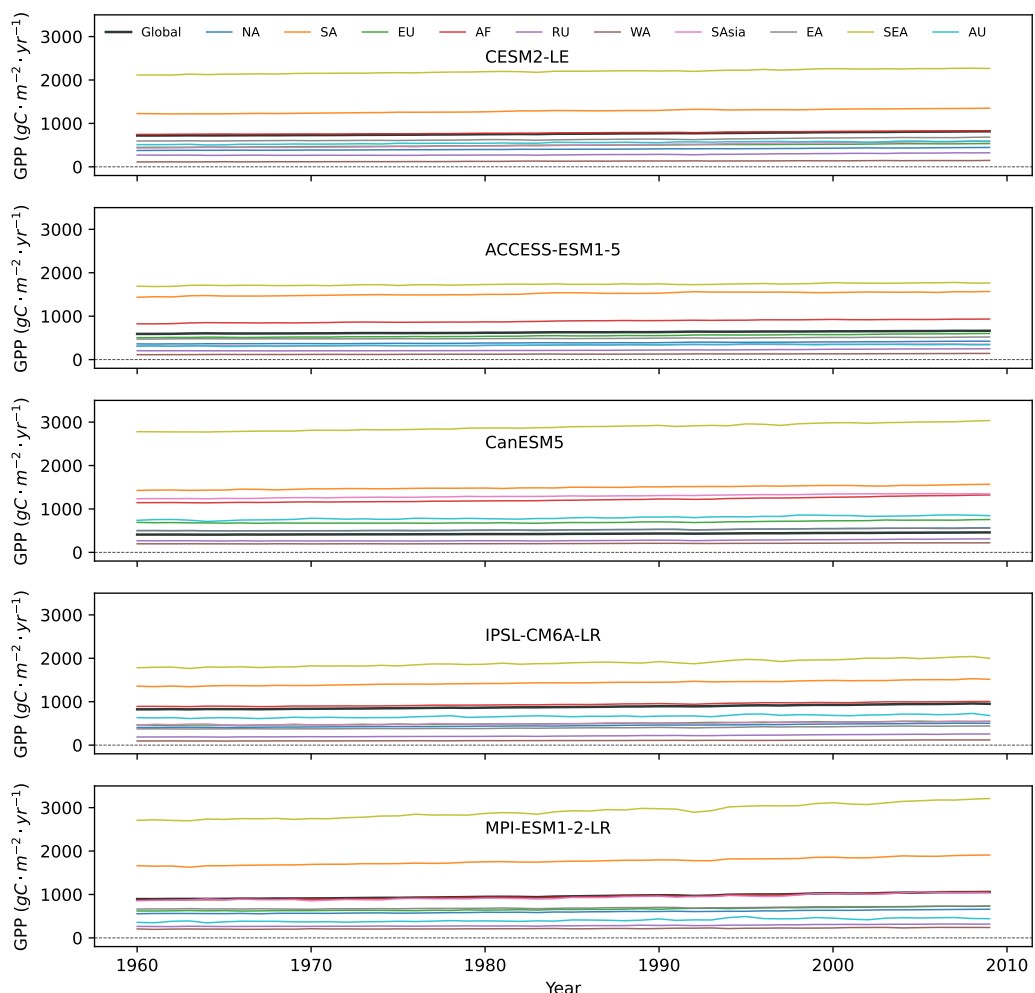

**Figure A.2.** Time series of GPP ensemble mean from five ESMs. The thick black line is the global ensemble mean, and the colored lines represent ensemble means for the 10 RECCAP-2 regions.

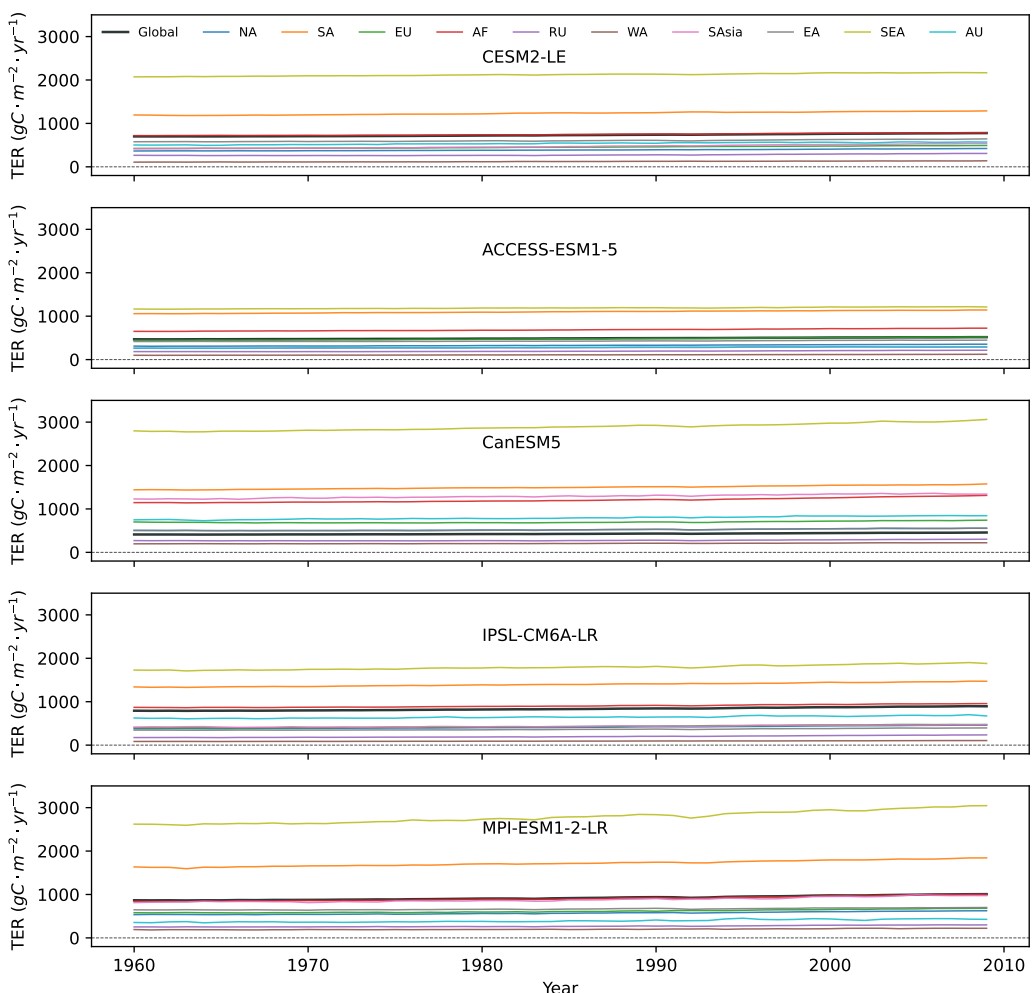

**Figure A.3.** Time series of TER ensemble mean from five ESMs. The thick black line is the global ensemble mean, and the colored lines represent ensemble means for the 10 RECCAP-2 regions.

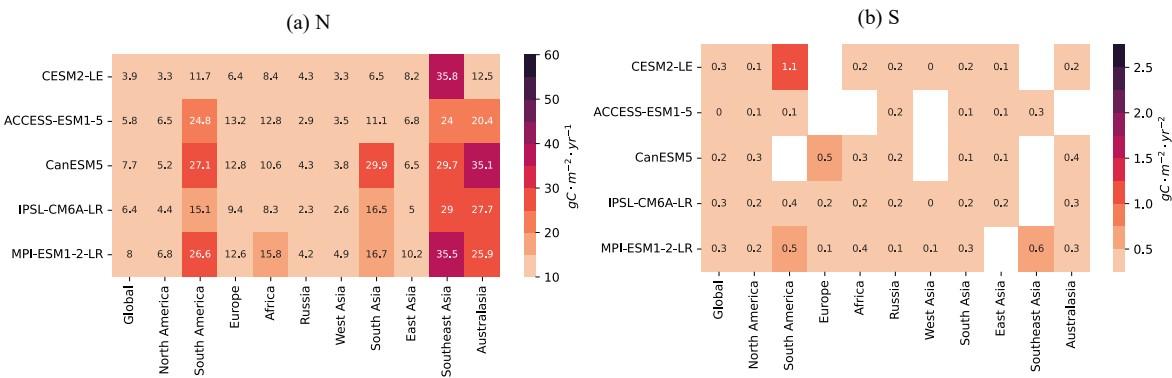

**Figure A.4.** Heat map of noise and signal of NBP in historical simulations across five ESM large ensembles.

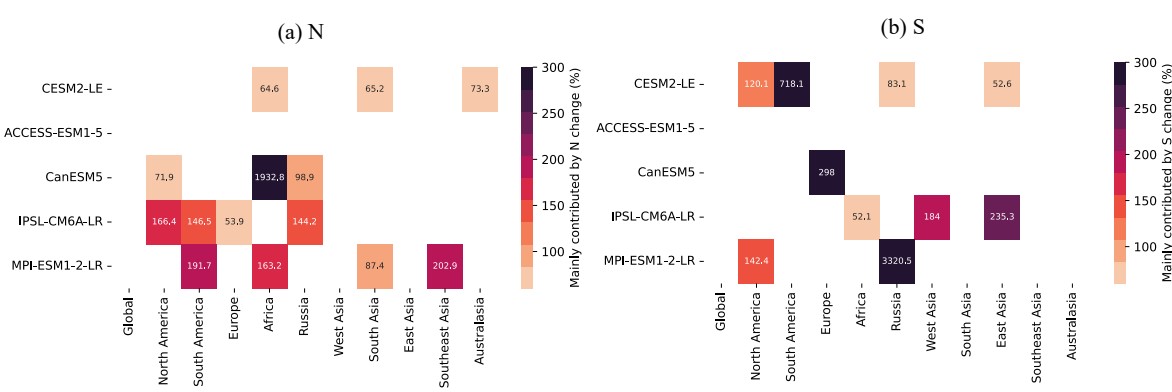

**Figure A.5.** Contribution of N and S to each RECCAP-2 region's ToE change in NBP, compared with global scale, in historical simulations. Note that we only show the values with N/S change as the dominant contributor.

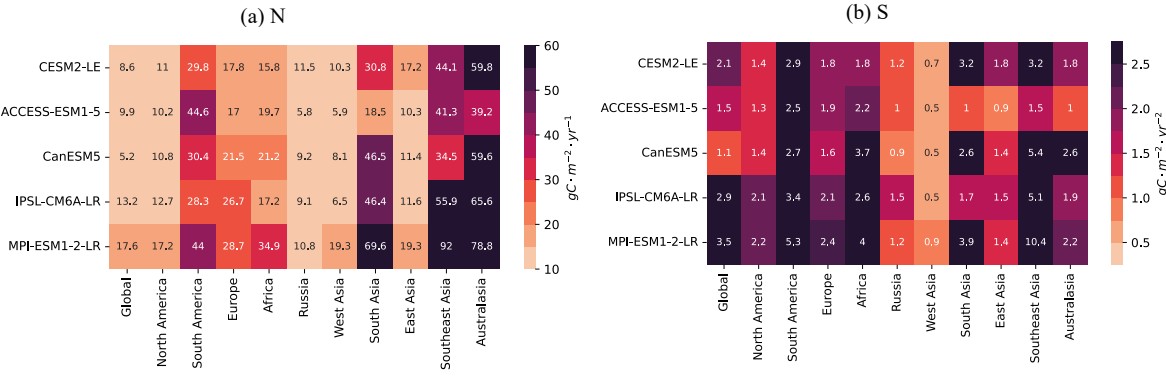

**Figure A.6.** Heat map of noise and signal of GPP in historical simulations across five ESM large ensembles.

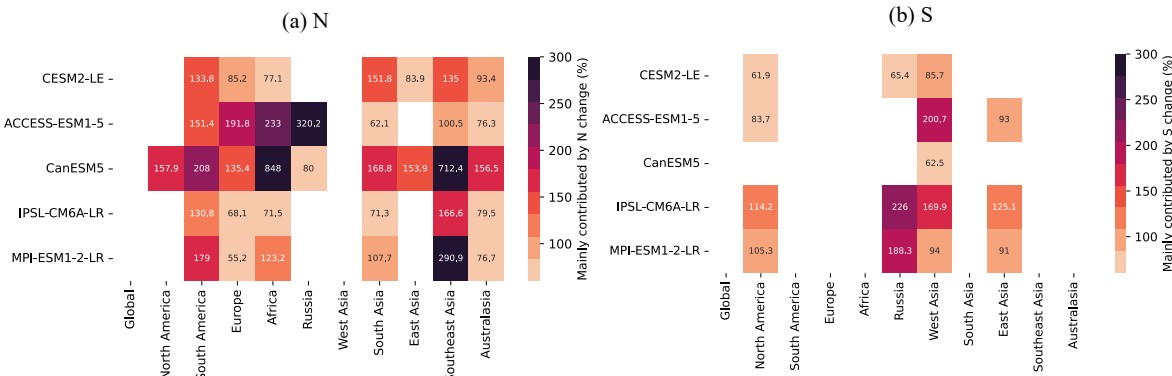

**Figure A.7.** Contribution of N and S to each RECCAP-2 region's ToE change in GPP, compared with global scale, in historical simulations. Note that we only show the values with N/S change as the dominant contributor.

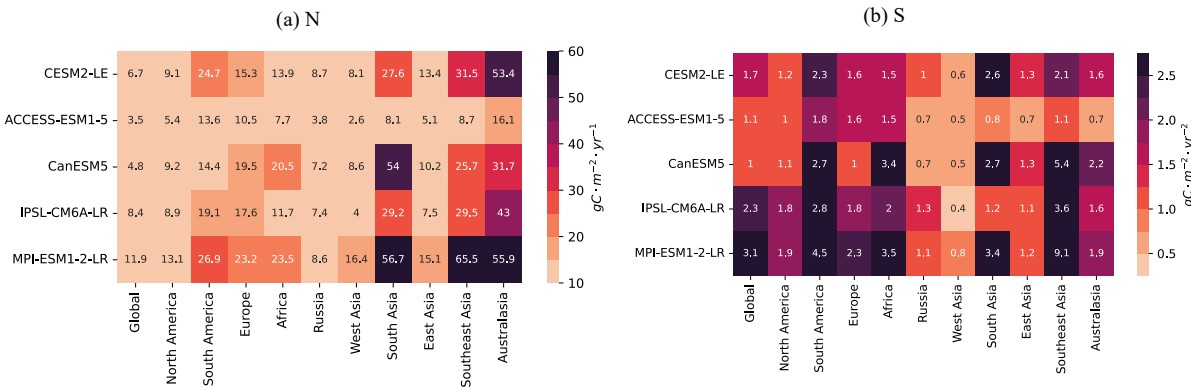

**Figure A.8.** Heat map of noise and signal in TER in historical simulations across five ESM large ensembles.

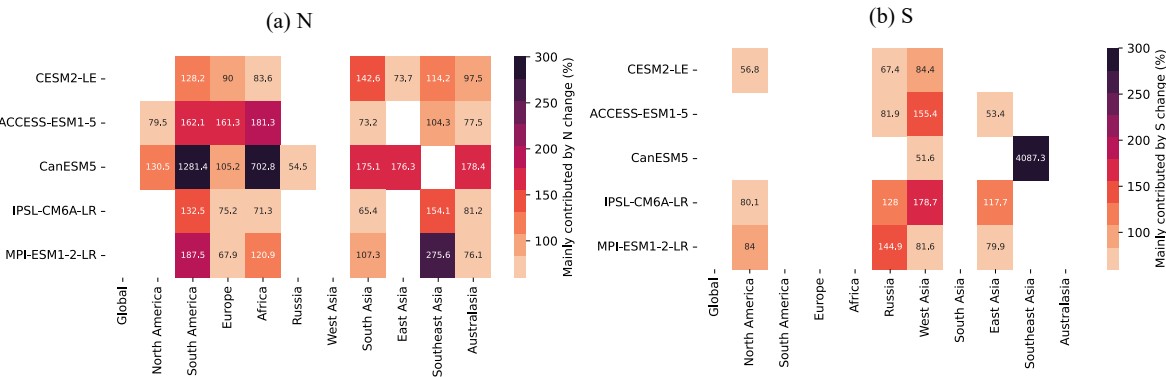

**Figure A.9.** Contribution of N and S to each RECCAP-2 region's ToE change in TER, compared with global scale, in historical simulations. Note that we only show the values with N/S change as the dominant contributor.

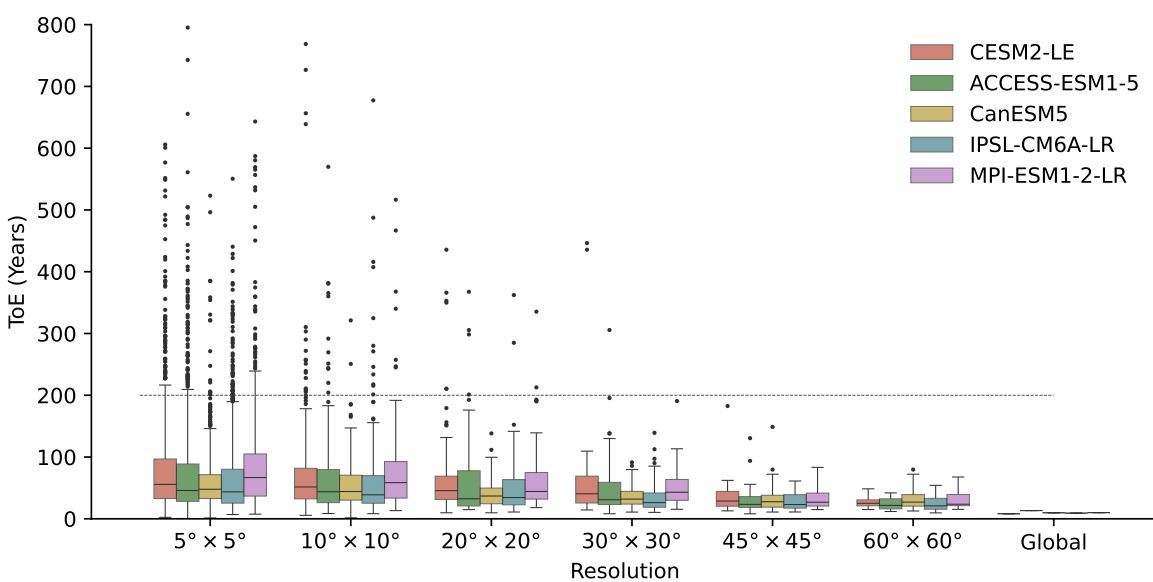

**Figure A.10.** Spatial effect in GPP historical simulations (1851–2014) across five ESM large ensembles. The distribution of time of emergence are shown for varying resolutions.

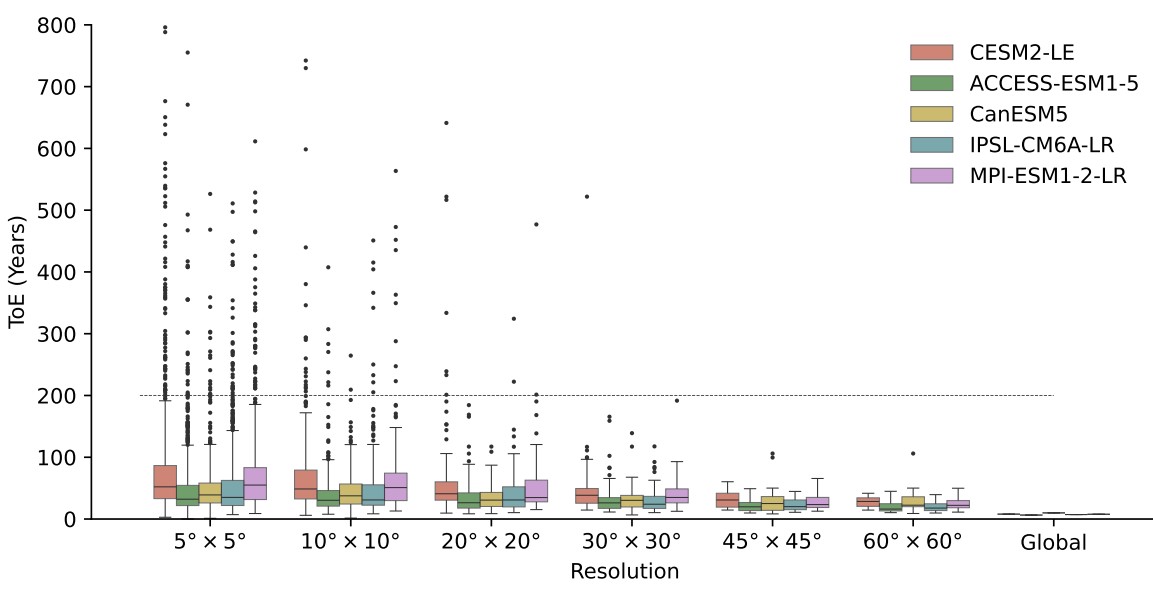

**Figure A.11.** Spatial effect in TER historical simulations (1851–2014) across five ESM large ensembles. The distribution of time of emergence are shown for varying resolutions.

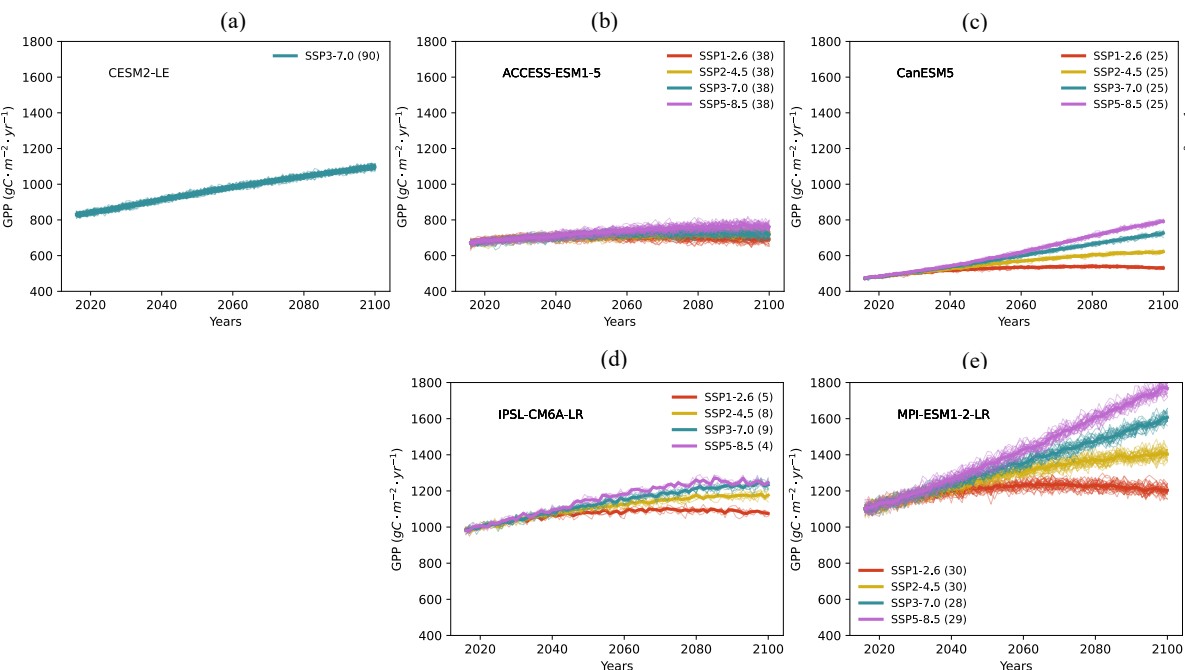

**Figure A.12.** The time series of future GPP from 2016 to 2100 across five ESM large ensembles. The four future scenarios include SSP1-2.6 (red line), SSP2-4.5 (yellow line), SSP3-7.0 (green line), and SSP5-8.5 (purple line). Thin lines represent individual simulations, while thick lines represent the ensemble mean for each scenario. The number of simulations for each model scenario is indicated in the legend next to the scenario label.

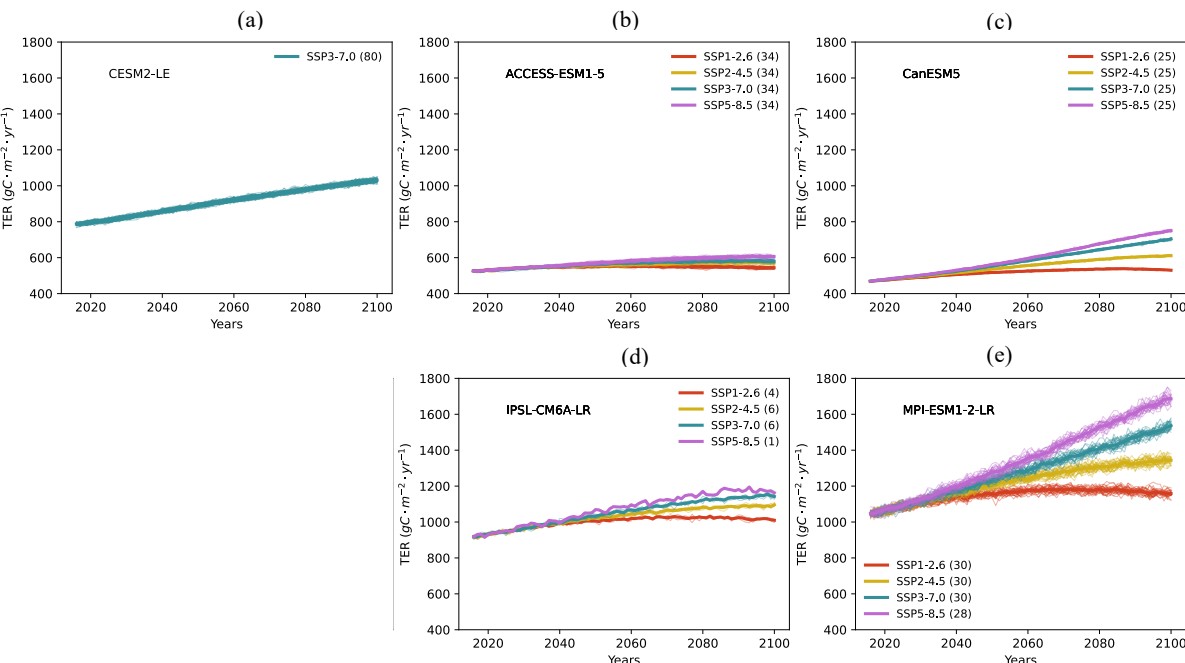

**Figure A.13.** The time series of future TER from 2016 to 2100 across five ESM large ensembles. The four future scenarios include SSP1-2.6 (red line), SSP2-4.5 (yellow line), SSP3-7.0 (green line), and SSP5-8.5 (purple line). Thin lines represent individual simulations, while thick lines represent the ensemble mean for each scenario. The number of simulations for each model scenario is indicated in the legend next to the scenario label.

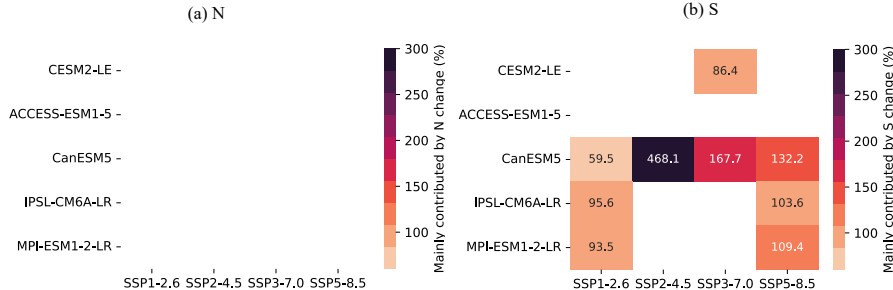

**Figure A.14.** Contribution of N and S to ToE changes in NBP for future scenario, compared with global scale in historical simulations. Note that only values where changes in N or S are the dominant contributor are shown.

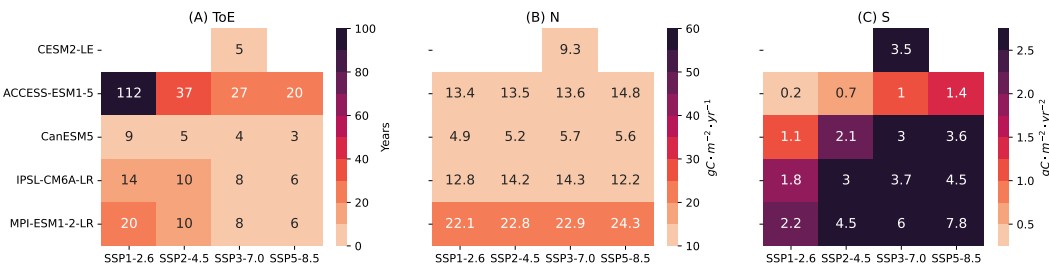

**Figure A.15.** Heat map of ToE, noise, and signal of GPP under future scenarios.

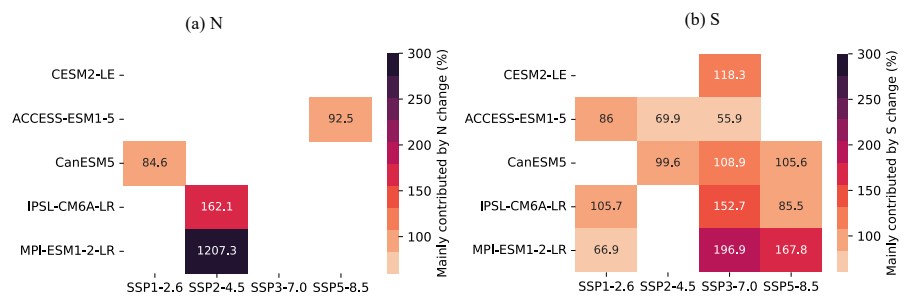

**Figure A.16.** Contribution of N and S to ToE changes in GPP for each future scenario, compared with global scale in historical simulations. Note that only values where changes in N or S are the dominant contributor are shown.

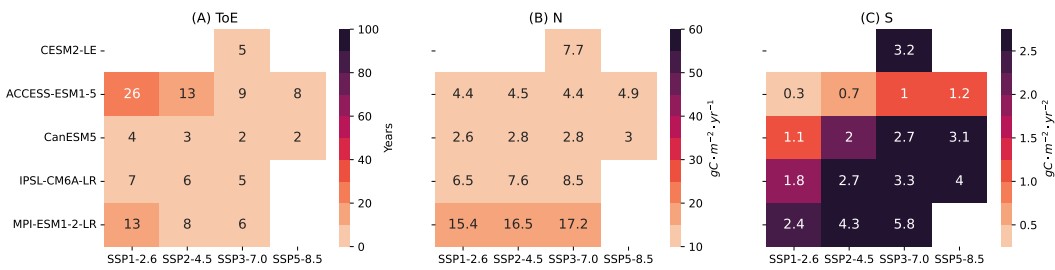

**Figure A.17.** Heat map of ToE, noise, and signal of TER under future scenarios.

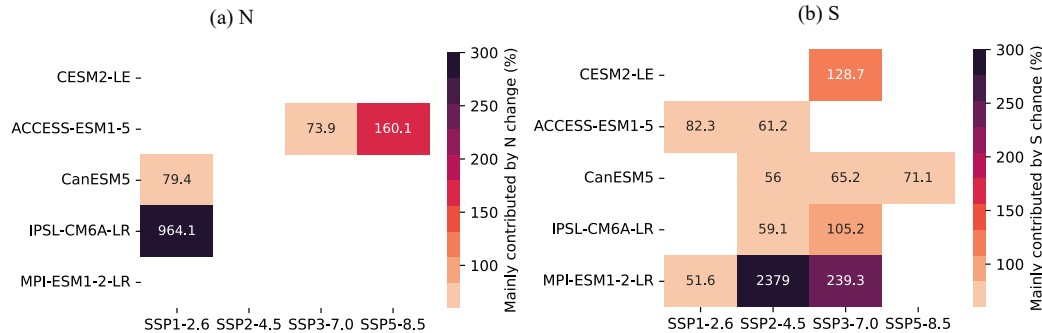

**Figure A.18.** Contribution of N and S to ToE changes in TER for each future scenario, compared with global scale in historical simulations. Note that only values where changes in N or S are the dominant contributor are shown.

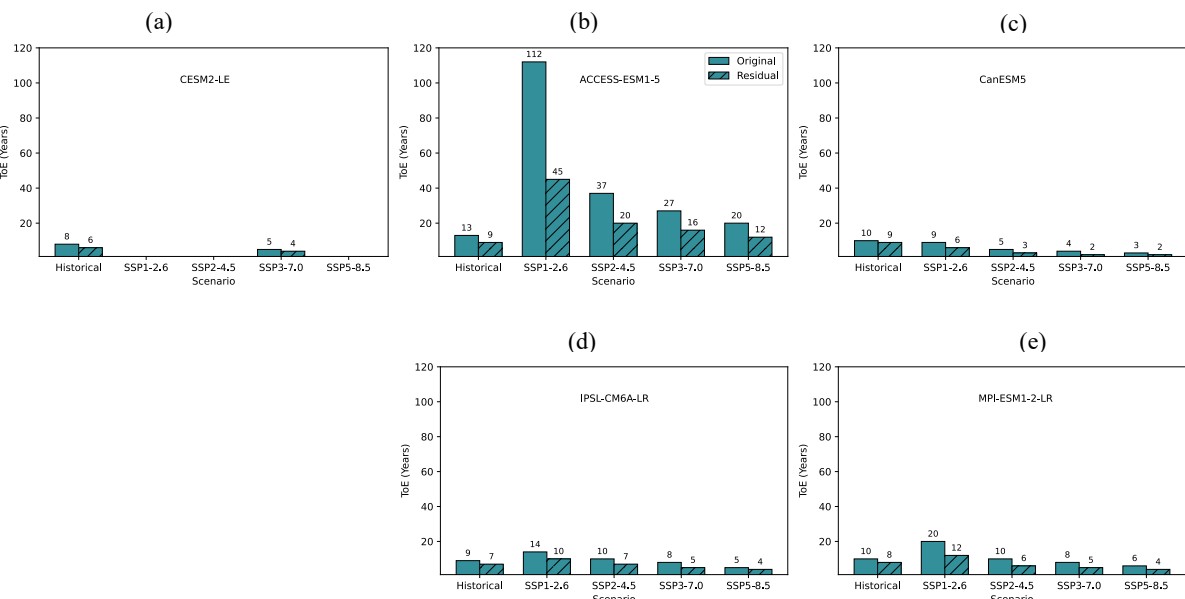

**Figure A.19.** ToE of GPP from historical simulations to future scenarios. The light colored boxes represent the ToE of GPP, while the neighboring darker shade, black framed boxes represent the ToE of the GPP residual, which has the circulation-induced variability removed.

| Relative change % (Years) | CESM2-LE | ACCESS-ESM1-5 | CanESM5 | IPSL-CM6A-LR | MPI-ESM1-2-LR |
| --- | --- | --- | --- | --- | --- |
| Historical | 34% (9) | - | 39% (26) | 35% (16) | 36% (19) |
| SSP1-2.6 | - | 2% (2) | 43% (64) | 34% (29) | 31% (30) |
| SSP2-4.5 | - | - | 53% (32) | - | - |
| SSP3-7.0 | 35% (15) | 34% (34) | 55% (19) | - | - |
| SSP5-8.5 | - | - | 52% (10) | 29% (42) | 37% (26) |

**Table A.1.** ToE reduction in NBP, calculated according to Eq. 4.

| Relative change % | CESM2-LE | ACCESS-ESM1-5 | CanESM5 | IPSL-CM6A-LR | MPI-ESM1-2-LR |
| --- | --- | --- | --- | --- | --- |
| Historical | 35% | 34% | 48% | 39% | 39% |
| SSP1-2.6 | - | 37% | 53% | 28% | 35% |
| SSP2-4.5 | - | 38% | 54% | 32% | 38% |
| SSP3-7.0 | 34% | 38% | 55% | 31% | 36% |
| SSP5-8.5 | - | 39% | 53% | 20% | 37% |

**Table A.2.** Noise reduction in NBP, calculated according to Eq. 4.

| Relative change % (Years) | CESM2-LE | ACCESS-ESM1-5 | CanESM5 | IPSL-CM6A-LR | MPI-ESM1-2-LR |
| --- | --- | --- | --- | --- | --- |
| Historical | 25% (2) | 32% (4) | 13% (1) | 26% (2) | 25% (2) |
| SSP1-2.6 | - | 60% (67) | 34% (3) | 26% (4) | 38% (7) |
| SSP2-4.5 | - | 45% (17) | 34% (2) | 28% (3) | 38% (4) |
| SSP3-7.0 | 21% (1) | 41% (11) | 41% (2) | 30% (2) | 36% (3) |
| SSP5-8.5 | - | 43% (9) | 34% (1) | 19% (1) | 37% (2) |

**Table A.3.** ToE reduction in GPP, calculated according to Eq. 4.

| Relative change % | CESM2-LE | ACCESS-ESM1-5 | CanESM5 | IPSL-CM6A-LR | MPI-ESM1-2-LR |
|---|---|---|---|---|---|
| Historical | 26% | 33% | 14% | 28% | 27% |
| SSP1-2.6 | - | 39% | 34% | 27% | 37% |
| SSP2-4.5 | - | 39% | 34% | 29% | 38% |
| SSP3-7.0 | 21% | 40% | 42% | 30% | 36% |
| SSP5-8.5 | - | 42% | 34% | 18% | 37% |

**Table A.4.** Noise reduction in GPP, calculated according to Eq. 4.

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
