# Peer review of "Quantifying the time of emergence of the anthropogenic signal in the global land carbon sink"

_EGUsphere, 2025_

## Author Comment (AC1)

**Reply to reviewers for the manuscript egusphere-2025-1924:"Constraining the time of emergence of anthropogenic signal in the global land carbon sink"**

**Na Li, Sebastian Sippel , Nora Linscheid , Miguel D. Mahecha , Markus Reichstein , and Ana Bastos**

We thank the reviewers for the productive and constructive comments. Here we reply to the comments in detail, and we suggest how to adjust the manuscript according to the comments. In addition, based on the comments, we made some extra changes in the manuscript for easier reading and understanding (at the last of each reply).

Below we address the changes according to the comments **(the line numbers in replies are according to the new manuscript that shows changes**). The blue colored lines are my replies, and the red words are locations changed.

**Changes according to the reviewer #1:**

**Specific comments:**

This paper considers the time of emergence of 3 key variables in the global land carbon sink and shows how dynamical adjustment can be used to shorten the detection time. While the paper is interesting and provides an important study in the literature I have a few major concerns around the methodology and lack of observational evidence. As such, I recommend major revisions.

We thank the reviewer for raising these important points, below we address the manuscript according to the comments.

1. The most major issue with the paper is the methods, which use linear trends to calculate ToE. This is not the typical method found in other studies such as: https://agupubs.onlinelibrary.wiley.com/doi/10.1029/2019GB006453 for large ensembles and https://agupubs.onlinelibrary.wiley.com/doi/10.1029/2011gl050087 for the standard method.

It is clear from the Figure time series that a linear fit is not appropriate for this data.

Additionally presenting ToE of 800 years, when the timeseries is only 100-years is confusing and extrapolation. If the signal has not emerged by the end of the timeseries I suggest presenting this as not emerged rather than extrapolating.

We acknowledge the reviewer's concern regarding the validity using a linear regression approach. Below we answer your concerns one by one:

      1)  Has the linear trend approach been applied before?

Yes, our approach is based on Bonan et al., 2021. Using fitted linear trend of ensemble mean as signal trend has been applied in recent published papers. The first paper mentioned (Schlunegger et al., 2020) also used fitted linear trend of ensemble mean as signal change, the similar approach as we did. They wrote (Schlunegger et al., 2020):

"For each LE, the signal is the ensemble average trend, computed as the average of the ~30 ensemble members' trends (linear, least squares trend)."

The second paper (Hawkins et al., 2012) mentioned used a nonlinear approach to represent signal trend: fitted fourth order polynomial regression of global mean SAT change. However, their study involves only around 10 simulations, so the ensemble mean might not reflect the full anthropogenic signal (Maher et al., 2019, Deser et al., 2020, Milinski et al., 2020). A fourth order polynomial might overfit the trend and miscapture some internal variability as signal trend, particularly in regional signal trend, which is subject to larger variability. In addition, their approach fits each simulation separately, so the fitted anthropogenic signals among all simulations might not be consistent within one single model. At last, the year to year calculation of ToE with actual fitted NBP value as signal might be more sensitive to anomalies.

To address the comment, we compared the Hawkins et al., (2012) with our method. We replicated almost exactly the steps described in Hawkins et al. (2012):

(i) For each Earth system model used in this study, we defined the signal and noise period of NBP in the same way as in our analysis: signal in the period 1960-2009 and noise 1930-1959.

(ii) At the global scale, for each model simulation, we fit a fourth order polynomial to calculate the fitted global signal trend of NBP. Noise for each model and each simulation is defined as the standard deviations of interannual variations.

(iii) At the regional scale, we used the 10 RECCAP domains. The signal trend for each domain is calculated as a scaled value of the corresponding global signal trend for each model simulation. The scaling procedure is applied exactly as in Hawkins et al. (2012):

$$S(t) = \alpha \tilde{T}_{\text{global}}(t) + \beta$$

S(t) represents the regional signal trend, and $T_{\text{global}}(t)$ is the fourth-order polynomial fitted over 1960-2009. Regional noise is defined as the standard deviations of interannual NBP of each model simulation.

(iiii) Time of Emergence (ToE) is defined as ToE = S/N. We identified the first year in which that ToE exceeded 2. This is equivalent to our definition, ToE = 2N/S.

(iiiii) For each model, we take the median ToE across all simulations.

For example, in the model CESM2-LE, we have 90 simulations. We define the signal period as 1960-2009 and the noise period as 1930-1959. On the global scale, we fit a four-order polynomial to each simulation to estimate the signal trend, with noise defined as the standard deviation. At the regional scale (e.g., North America), the domain signal trend is scaled to the corresponding global trend, and noise is the standard deviations within each domain. For each simulation, ToE is calculated year by year, and the first year in which ToE exceeded 2 was recorded. The median ToE across the 90 simulations is reported as the final emergence time.

Below we show (Fig. S1) the ToE for NBP (global and 10 RECCAP domains) using the approach in Hawkins et al. (2012):

[Figure]

Figure S1. ToE of global and 10 RECCAP domain NBP. ToE is calculated according to the approach in Hawkins et al. (2020). ToE for each model each region is the median of all simulations.

We compare this with our approach in Figure 2 Panel b of our manuscript.

[Figure]

Figure 2: ToE of NBP on a global scale and across 10 RECCAP-2 regions, under historical simulations of five ESM large ensembles. Note that ToE is the years detectable after 1960, and is calculated with signal period of 1960-2009 relative to the noise period of 1930-1959, details please check Sect. 2.4. Heat map of the ToE in global and each spatial domain of NBP. Domains with no significant signal (P > 0.05) are shown as empty squares.

We found the Hawkins et al. (2012) approach (Fig. S1) generally yields shorter ToEs than our linear approach (Fig. 2). However, in their approach, two regions show a ToE of only one year (CESM2-LE Europe and ACCESS Russia). This might be due to large signal values in 1960, or small noise in the period 1930-1959. For example, CESM2-LE

Europe with a median ToE of 1 year, across all 90 ensemble simulations. The ensemble mean since 1960 is apparently larger than most other 10 RECCAP domains (Fig. A1 in the manuscript shown below, a panel of ensemble mean as suggested by the reviewer). There might be an elevated value of fitted NBP signal trend since 1960, across most simulations in this region. Thus their approach is sensitive to the reference period selected and year by year ToE calculation makes their approach more sensitive to anomalies. Instead, we calculate the signal trend of the European ensemble mean in the period 1960-2009, the fitted linear slope shows no significant trend in this period (see Fig. A4 in the manuscript). Thus the emergence time is not detectable in this region.

[Figure]

Figure A.1. Time series of NBP ensemble mean from five Earth system models. The thick black line is the global ensemble mean, and the colored lines represent ensemble means for the 10 RECCAP-2 regions.

Because ToE in the nonlinear approach is defined as the median across simulations, the exact signal or noise value cannot be directly identified. Instead, we calculated the number of simulations in each model with ToE >2 (Fig. S2).

[Figure]

Figure S2. In the Hawkins et al (2012) approach, the number of simulations in each domain that show ToE > 2. The number of simulations included in this study are listed after each model name.

Fig. S2 shows that in many regions there are fewer than 10 simulations that exceed this threshold, meaning that less than one-third of simulations detect emergence. This indicates that the anthropogenic signals captured across simulations are not consistent.

In summary, although the nonlinear approach of Hawkins et al. (2012) generally produces shorter ToEs than our approach, it is less robust. Their approach shows: 1) unrealistic short detection times (1 year); and 2) in most regions, more than two-thirds of simulations fail to detect the anthropogenic signal. 3) the anthropogenic signals captured across models are not consistent.

This might be due to: instead of defining the signal trend as the ensemble mean, Hawkins et al. (2012) defined the signal trend separately for each simulation and then took the medians across simulations. This makes the signal trends vulnerable to anomalies in individual simulations, blurring the separation between anthropogenic signal and internal variability. Furthermore, with fewer than one-third of simulations detecting the signal, the reported median of ToEs are less reliable. Their method also calculates ToE year by year (e.g., one year ToE in Fig. A1), whereas we calculate the signal trend over the entire signal period.

This shows that the nonlinear approach used by Hawkins et al. (2012) is less robust on capturing the consistent anthropogenic signal and more sensitive to anomalies, particularly with net land carbon sink time-series, which is subject to large interannual variations.

    2) Is linear fit for signal trend appropriate in this study?

Yes. As concerns about the validity of linear regression slope, rather than a nonlinear approach, to represent the signal trend, we have below main reason:

Capturing the dominant signal (first-order degree variability). The ensemble mean reflects the forced signal in the Earth system, including anthropogenic forcing (both

long-term consistent and short-term surges), short period natural forcings (e.g., volcanic eruptions), and even decadal internal variability (Deser et al., 2012b; Canadell et al., 2021; Eyring et al., 2021; Mercado et al., 2009; Zhang et al., 2021). The linear fitted trend of the ensemble mean captures the most consistent and incremental influence of the anthropogenic signal, while minor deviations from the linear trend slope may be ignored. A nonlinear regression might capture these minor fluctuations but at the risk of overfitting and misattributing natural forcing or decal internal variability to anthropogenic signals. This risk is particularly pronounced at regional scales in the carbon cycle, where the ensemble mean exhibits larger variability (see Figure A.1 and Figure 3). Additionally, ToE comparisons across models is better possible with linear trends with two consistently interpretable parameters. In addition, the linearized trend is only applied over five decades (not over the whole period), and for a limited time period the linearization of the trends is more justified than for the whole period.

We have added one small paragraph explaining why linear regression is suitable for this purpose. In line 140-145:

"Here we use a linear regression slope rather than a nonlinear approach to represent the signal trend, this is to capture the dominant forced signal in the selected signal period. The ensemble mean of NBP, GPP and TER reflects the forced ecosystem response, including anthropogenic forcing, short-period natural forcings (e.g., volcanic eruptions), and decadal internal variability (Deser et al., 2012b; Canadell et al., 2021; Eyring et al., 2021; Mercado et al., 2009; Zhang et al., 2021). The linear trend captures the first-order long-term anthropogenic influence, whereas nonlinear methods could risk overfitting and misattributing natural or internal variability to anthropogenic signals, especially at regional scales where variability is larger (see Figure A.1 and Figure 3)."

3) Why include ToEs beyond the years of the selected study period?

We thank the reviewer for this question. Including Time of Emergence (ToE) values that extend beyond the study period provides important information about regions which might be dominated by natural variability, or are weakly influenced by anthropogenic forcing. The timing of these extended ToEs indicate how severe such limitations are in particular regions and climate situations. Since most future scenarios show ToEs below 150 years, and after reducing the circulation-induced noise in NBP, the ToEs show apparent reduced years. Here, to show the filter effect by using Ridge regression, we cut the ToEs to 150 years.

Additionally, we add this information in line 188-190:

"Note that only the calculated signal (regression slope) with significance value $P < 0.05$ is selected. If the calculated signal (regression slope) is negative, we then take the absolute signal value to get a positive ToE. Here we select to show the ToEs less than 150 years."

2. This method is used typically to understand when we might expect to observe changes outside the noise. However, the paper has no observational evidence in it. I suggest comparing to any observations we have.

We thank the reviewer for pointing out this critical issue, we added the evidence of using observations below, also as Figure 7:

[Figure]

Figure 7. Time series of the atmospheric $CO_2$ growth rate AGR at Mauna Loa from 1960 to 2009 (Lan et al., 2025). Five volcanic years (1963, 1982, 1983, 1991, and 1992) are removed. The red line is the observed AGR. The black line is the long-term trend fitted with a locally weighted scatterplot smoothing (Cleveland et al., 1991, LOWESS) (signal). The residual (AGR - fitted long-term trend) was predicted using SLP through ridge regression with leave-one-out cross validation (blue line). This SLP predicted residual is then subtracted from the observed AGR to obtain a new AGR time series with circulation-induced variations removed (observed AGR - SLP predicted residual). The dashed black line is the new long-term trend. Data pretreatment and the ridge regression model follow paper Li et al. (2021). Note that the signal period is the same as in models (1960—2009). Due to limited records of $CO_2$ observations before 1958, here we calculate the noise also in the period 1960—2009."

Accordingly, we add a paragraph to line 328 :

"We then test the same approach on observations of Atmospheric $CO_2$ growth rate (AGR) from Mauna Loa (Lan et al., 2025) for the period 1960–2009, matching the period of signal for the Earth system model analysis (Fig. 7). The ToE of the observed AGR is 33 years, with noise value of 0.87 gC· $yr^{-1}$ and signal value of 0.05 gC· $yr^{-2}$ (Fig. 7). After removing circulation-induced variations through dynamical adjustment, the ToE of the adjusted new AGR is reduced to 23 years, with noise of 0.70 gC· $yr^{-1}$ and signal of 0.06 gC· $yr^{-2}$ (Fig. 7). This represents an overall reduction of about 30%, contributed by 19% reduction in noise and 20% increase in signal. The results show that this approach can be applied in observations, enabling earlier detection of anthropogenic signals in global carbon cycle variability."

We also added the information to abstract,  Methods section "Data" and "Data preatment":

In line abstract line 17-19:

"This approach substantially shortens the detection time for the global net land carbon sink: between 34–39% for the historical period and 27–54% for the future simulations. This approach  can also shorten the detection time for observational based datasets (30% reduction), thereby improving our ability to detect long-term trends of land carbon sink variability."

In line 111-112:

"We also included the observations of atmospheric $CO_2$ growth rate (AGR) at Mauna Loa from 1960 to 2009 (Lan et al., 2025), downloaded from https://gml.noaa.gov/webdata/ccgg/trends/co2/co2_gr_gl.txt, last accessed on August 18th."

And line 125-127:

"The pretreatment steps of atmospheric $CO_2$ growth rate (AGR) at Mauna Loa from 1960 to 2009 (Lan et al., 2025) follows Li et al. (2022). We first have five volcanic years (1963, 1982, 1983, 1991, and 1992) removed, then remove the long-term trend by fitting with a locally weighted scatterplot smoothing (Cleveland et al., 1991, LOWESS)."

3. The % reductions from the dynamical adjustment might be good to include as years as well as % as this is easier to understand. For the dynamical adjustment it is unclear whether this method could be applied to observations to reduce the TOE or just models. I suggest adding this information in the text.

Thanks for pointing out, we added years that are equivalent to % reduction in Table A.1 and A.3. However, it should be noted that higher relative reduction % do not imply longer year reductions among models and study periods. Table A.1 and A.3 now looks like below (note that we changed the noise period from 2020-2050 to 2020-2070, so the values below are slightly different):

| Relative change % (Years) | CESM2-LE | ACCESS-ESM1-5 | CanESM5 | IPSL-CM6A-LR | MPI-ESM1-2-LR |
|---|---|---|---|---|---|
| Historical | 34% (9) | - | 39% (26) | 35% (16) | 36% (19) |
| SSP1-2.6 | - | 2% (2) | 43% (64) | 34% (29) | 31% (30) |
| SSP2-4.5 | - | - | 53% (32) | - | - |
| SSP3-7.0 | 35% (15) | 34% (34) | 55% (19) | - | - |
| SSP5-8.5 | - | - | 52% (10) | 29% (42) | 37% (26) |

**Table A.1.** ToE reduction in NBP, calculated according to Eq. (4).

| Relative change % (Years) | CESM2-LE | ACCESS-ESM1-5 | CanESM5 | IPSL-CM6A-LR | MPI-ESM1-2-LR |
|---|---|---|---|---|---|
| Historical | 25% (2) | 32% (4) | 13% (1) | 26% (2) | 25% (2) |
| SSP1-2.6 | - | 60% (67) | 34% (3) | 26% (4) | 38% (7) |
| SSP2-4.5 | - | 45% (17) | 34% (2) | 28% (3) | 38% (4) |
| SSP3-7.0 | 21% (1) | 41% (11) | 41% (2) | 30% (2) | 36% (3) |
| SSP5-8.5 | - | 43% (9) | 34% (1) | 19% (1) | 37% (2) |

**Table A.3.** ToE reduction in GPP, calculated according to Eq. (4).

We also added the equivalent years to % reduction in line 320:

"ToE ranges from 34% (CESM2-LE) to 39% (CanESM5), corresponding to 9 and 26 years, respectively."

Line 322:

" reduction ranges from 29% to 55% (42 and 19 years reduction, respectively)"

Line 324:

"it ranges from 13% (CanESM5) to 32.% (ACCESS-ESM1-5), corresponding to 1 and 4 years, respectively. For future scenarios, the relative reduction ranges from 19.% to 60% (1 and 67 years reduction, respectively)."

We also added experiment results of observations as Figure. 7 (as shown above).

4. I am confused about the use of 2020-2050 for N and 2020-2070 for S. What makes this choice? This seems very arbitrary

For the signal period, the trend of S after 2020 is more sensitive to the period selected. Across the four future scenarios, NBP shows much larger variations than in the historical runs, and almost all the signals are mixed before 2050. In addition, all four future scenarios define key $CO_2$ emission milestones by 2050 (see section 2.1). To reduce the influence of large annual variability (by selecting a period of at least 30 years) and to fully capture the impact of these policy milestones, we selected 2070 as the end year. According to Figure 4, the standard deviation of N is not sensitive to the specific period selected. We therefore choose 2020-2070, the same period with signal. The results have minor changes accordingly.

5. Figures

Figure 1: You could easily include all NBP, GPP and TER on one figure – I would recommend this rather than push some to the supplementary. I also suggest adding a multi-ensemble mean panel. The ensemble sizes seem rather arbitrary. For example ACCESS has 40 members and

CESM2 100 members – so why are they only 38 and 90? Additionally CESM2 biomass burning runs have been shown to be different to the cmip6 runs have you checked these differences?

1. Include all NBP, GPP, and Ter in one figure.

Thanks, we combined NBP, GPP and TER in Figure 1.

[Figure]

Figure 1. Time series of NBP, GPP, and TER from 1851 to 2014 in five ESM large ensembles. The thin lines represent individual simulations, while the bold lines represent the ensemble mean. The gray lines are NBP follows the left y-axis. The blue and red lines follow the right y-axis and are GPP and TER, respectively. The number of simulations for each model is listed in the legend next to the model name. Note that TER in model ACCESS-ESM1-5 only included 24 simulations, due to limited data availability.

2. Adding a multiple ensemble panel

We included three figures of multi-ensemble mean (global scale and 10 RECCAP-2 Regions) in supplementary Figure A.1 to A.3, for NBP, GPP, and TER respectively. Below are three Figures:

[Figure]

Figure A.1. Time series of NBP ensemble mean from five Earth system models. The thick black line is the global ensemble mean, and the colored lines represent ensemble means for the 10 RECCAP-2 regions.

[Figure]

Figure A.2. Time series of GPP ensemble mean from five Earth system models. The thick black line is the global ensemble mean, and the colored lines represent ensemble means for the 10 RECCAP-2 regions.

[Figure]

Figure A.3. Time series of TER ensemble mean from five Earth system models. The thick black line is the global ensemble mean, and the colored lines represent ensemble means for the 10 RECCAP-2 regions.

Accordingly, we add two sentences to line 221:

"We first check the ensemble mean of NBP in global scale and 10 RECCAP-2 regions (Appendix A Fig. A1). The trends of ensemble mean in NBP subjects to larger variability in regional than in global scales, particularly in Southeast Asia and Africa."

3. The ensemble size looks arbitrary.

The four CMIP6 simulations are originally retrieved from https://esgf-node.llnl.gov/projects/cmip6/, and pre-processed by Brunner et. al. (2020), including aggregate to resolution of 2.5° × 2.5° and resampled to annual mean. The number of simulations for each model from pre-processed data from Brunner et al. (2020). CESM2-LE is downloaded from https://rda.ucar.edu/datasets/d651056/#. They include 90 simulations of NBP. We agree with the reviewer: The different large ensembles come in different sample sizes and therefore look somewhat arbitrary, as they are provided by different modeling centers who make their own decision regarding sample sizes; yet this is unfortunately nothing we could change or streamline for the paper.

4. CESM2-LE burned area influenced ensemble mean?

CESM2-LE includes 90 simulations: the first 50 are CMIP6 simulations and the remaining 40 are SMBB (biomass burning) simulations. We compared the ensemble mean of the first 50 CMIP6 simulations with that of the 40 SMBB simulations, and found the difference in global scale negligible. The figure below shows that the two types of simulations are well mixed, with similar ensembles means.

[Figure]

Figure 2:

Suggest adding a multi-ensemble mean to each panel – this suggestion is for all Figures. For this figure is it showing TOE = years post 1960? can this be clear in the caption please. Additionally what does 'the spatial domains are slightly changed' mean?

We included three figures of multi-ensemble mean (global scale and 10 RECCAP-2 Regions) to supplementary Figure A.1 to A.3, for NBP, GPP, and TER respectively. As is shown above.

Figure 2 shows ToE years after 1960. We added this to the caption:

"Figure 2. ToE of NBP on a global scale and across 10 RECCAP-2 regions, under historical simulations of five ESM large ensembles. Note that ToE is the years detectable after 1960, and is calculated with signal period of 1960-2009… "

"The spatial domain are slightly changed" means that after we aggregated the RECCAP-2 map from 0.5°×0.5° to 2.5°×2.5°, the aggregated map looks a bit different as in panel a, Figure 2.

Figure 3 – Is this really resolution? i.e. is it the same area regridded to a different resolution or is it a different spatial domain?

Yes, they are all spatial resolutions. We aggregated the global data per pixel to different resolutions, then calculated the ToE per pixel. Figure 3 shows the ToEs of all global pixels.

We add these explanations to the caption of Figure 3:

"Spatial effect in NBP historical simulations across five ESM large ensembles. The distribution of ToE (years after 1960) is shown for varying spatial resolutions. We aggregate the global data per pixel to different resolutions, then calculate the ToE per pixel. The line within each box indicates the median. Note that all signals are in absolute values, so the calculated ToE are all positive."

Figure 5: there seem to be missing bars where only the light or shaded box is there? What happened to the other box?

As we stated in method section 2.4:

"Note that only the calculated signal (regression slope) with significance value P < 0.05 is selected."

We only present the ToE with signal significance P < 0.05. The missing bars have a signal that is not significant (P > 0.05), which implies the linear slope (signal trend) has not emerged, so we did not include them.

We explained this further in the caption of Figure 6 (old Figure 5):

"In cases where both boxes are missing, the respective signal is not available (no significance of linear trend slope), or the ToEs are longer than 150 years. ."

A lot of content discussed is in the appendix figures – I wonder if more should be in the main text

We added Figure A.9 (Heat map of ToE, noise, and signal of NBP under future scenarios.) up as new Figure 5. Note that since we changed the noise period from 2020-2050 to 2020-2070, so the new Figure 5 with slightly different values.

[Figure]

Figure 5. Heat map of ToE, noise, and signal of NBP under future scenarios.

6. Paper is missing a discussion section

Thanks for pointing out, we discussed mostly at the end of each section and also in conclusions. Here, we added a formal discussion:

In line 260:

"We found global scale takes shorter time to detect long-term trends induced by anthropogenic effects than at smaller scales, with ToE increasing for smaller domains as reported by Lombardozzi et al. (2014), though their study used fewer models and less than 10 simulations.  For regional NBP, except large internal variability, the larger variability of the signal trend also contributes, likely reflecting different regional climate drivers (e.g., fires, decadal internal variability, land use changes (Deser et al., 2012b; Canadell et al., 2021; Eyring et al., 2021; Mercado et al., 2009). A few regions, however, show shorter ToEs than the global scale. For example, in Russia, CanESM5 and IPSL-CM6A-LR simulate relatively small noise and stronger signal trends, leading to shorter ToEs. This maybe relate to the sparsely distributed ecosystems in Russia, which are less sensitive to changes in climate drivers.

It should also be noted that the large interannual variations in NBP largely arise from variability in GPP and respiration. Future studies can focus on comparing regional GPP and respiration variabilities and attributing them to different drivers. Regional ecosystems are more influenced by precipitation than by temperature (Jung et al., 2017), implying that large variations in regional NBP maybe driven by precipitation. However, the detection of anthropogenic signals in precipitation is less robust and delayed than in temperature (Doblas-Reyes et al., 2021). Thus, detecting anthropogenic signals in precipitation and assessing how they influence regional ecosystem activity, may provide a cleaner and earlier detection of anthropogenic signals in regional land carbon sinks."

In line 298:

"The divergence of NBP across models is much larger in future scenarios than in historical simulations. While models are inherently different, these differences in the historical period may be amplified in future scenarios due to: (a) the increasing influence of $CO_2$ fertilization and land use change (van Vuuren et al., 2011, Ciais et al., 2013, O'Neil et al., 2016, Christensen et al., 2018, Arora et al., 2020, Lee et al., 2021); and (b). Rising temperature and more frequent extreme events, which may intensify differences in climate-carbon feedbacks among models (Friedlingstein et al., 2014, Hewitt et al., 2016, Seneviratne et al., 2021)."

We also added one paragraph in conclusion (the last second paragraph), in line 368:

"The emergence approach used in this study is sensitive to the choice of the reference period, and signal and noise length. In addition, the signal trend can be biased by the large regional variability, and the noise can enhance or amplify external natural forcing in

observations (Lombardozzi et al., 2014, Bonan et al., 2021). A better understanding of regional ecosystem responses to anthropogenic signals, and improved methods may help improve the detected emergence time."

7. Section 3.1 – the noise could be easily quantified rather than discussed qualitatively

Thanks for pointing out this very critical issue.

1) we quantified the percentage of the noise change that contributed to the ToE change in the manuscript, both changes are compared with global scale. We added the figures according to the manuscript as Figure A.5, A.7, and A.9 for historical RECCAP-2 region contribution (compared with global scale), as well as Figure for future contribution (compared with historical global scale).

2) We added one paragraph in section 2.4 (line 194) to explain the methods

3) We rewrote some parts in Section 3.2 and 3.3 (line 221-290), as well as the conclusion to include the quantified N and S contributions.

Below are details:

1)  Quantify N and S contribution

   Due to ToE = 2N * (1/S), both N and S have changed in regional scale, compared with global scale.

   ToE_global = 2N_global * (1/S_global)

   ToE_region = 2N_region * (1/S_region)

   We calculate the contribution of N and S by taking the Log of two sides of the equation.

   log(ToE_global) = log(2N_global) + log(1/S_global) (1)

   log(ToE_region) = log(2N_region) + log(1/S_region) (2)

   Using equation 2 minus equation 1.

   log(ToE_region) - log(ToE_global) = log(2N_region) - log(2N_global)

   + log(1/S_region) - log(1/S_global)   (3)

   Then we can calculate the percentage of log changes in the noise and in the signal that contribute to log ToE change.

   Below is the calculated percentage of contribution from N and S in historical simulations, here we only show the values with N/S change as the dominant contributor.

[Figure]

Figure A5. Contribution of N and S to ToE changes in NBP for each RECCAP-2 region, compared with global scale, in historical simulations. Note that only values where changes in N or S are the dominant contributor are shown.

[Figure]

Figure A7. Contribution of N and S to ToE changes in GPP for each RECCAP-2 region, compared with global scale, in historical simulations. Note that only values where changes in N or S are the dominant contributor are shown.

[Figure]

Figure A9. Contribution of N and S to ToE changes in TER for each RECCAP-2 region, compared with global scale, in historical simulations. Note that only values where changes in N or S are the dominant contributor are shown.

[Figure]

Figure A14. Contribution of N and S to ToE changes in NBP under each future scenario, compared with the historical global scale. Note that only values where changes in N or S are the dominant contributor are shown.

[Figure]

Figure A16. Contribution of N and S to ToE changes in GPP under each future scenario, compared with the historical global scale. Note that only values where changes in N or S are the dominant contributor are shown.

[Figure]

Figure A18. Contribution of N and S to ToE changes in TER under each future scenario, compared with the historical global scale. Note that only values where changes in N or S are the dominant contributor are shown.

2) Add a paragraph to Section 2.4, describing the method used to calculate the contribution of changes in N and S to changes in ToE. In line 194:

"We also calculated the contribution of N and S to ToE changes in each RECCAP-2 region, compared with the global scale, for NBP, GPP, and TER. The Eq. (5) is the logarithmic (in base e) form of Eq. (3). We first calculate the log change in global and regional scales. Then calculate the differences between each region to global scale (Eq. (6)).

$$\ln(ToE) = \ln(2N) + \ln(1/S) \quad (5)$$

$$\ln(ToE_{region}) - \ln(ToE_{global}) = \ln(2N_{region}) - \ln(2N_{global}) + \ln(1/S_{region}) - \ln(1/S_{global}) \quad (6)$$

The contribution of changes in N and S are:

$$N_{contri} = 100\% * (\ln(2N_{region}) - \ln(2N_{global})) / (\ln(ToE_{region}) - \ln(ToE_{global})) \quad (7)$$

[revised manuscript text omitted]

Line 376: "This approach can be has been applied in observations to facilitate and shows an early detection of anthropogenic signal in global carbon cycle variability."

**Minor comments**

Title should be 'the anthropogenic signal'

Thanks, we added "the" to the title.

**Intro**

Line 1 'the' atmospheric CO2

Thanks, we added "the" to line 1: "..., driven by the increasing atmospheric $CO_2$…", and also in line 24: "...mainly driven by the increasing atmospheric $CO_2$…"

Line 2 – this is not driven by climate change, more likely it is driven by processes related to climate change please be specific here

Thanks for pointing out this, we rewrite line 2 as: "...driven by the increasing atmospheric CO2 concentration and physical processes influenced by climate change."

line 6 replace by with from

We have replaced "by" with "from" in line 6: "and future scenarios (2016-2020)  from Earth system models."

Line 13 – it is odd to have a 1 sentence paragraph – this sentence is also unclear can you please rephrase

Thanks, we have rephrased in line 13 as below:

"Secondly, future scenarios show delayed signal detection compared to historical trends.  This delay is mainly due to weaker anthropogenic signal trends rather than stronger natural variability. The weaker signal reflects the slow-down of the increasing net land carbon sink in response to emission mitigation."

Line 16 – remove 'allows to' replace shorten with 'shortens'

We changed line 17: "This approach  substantially shortens the detection time "

Line 18 – replace on with 'of'

We changed in line 19: "our ability to detect long-term trends  of land carbon sink variability"

Line 30 – should be 'temperature, and precipitation'

We changed line 30: "such as temperature and precipitation"

Line 33 – replace by with 'on'

Thanks we replaced in line 34: "The long-term trends of the global carbon cycle are superimposed  with substantial year–to–year".

Line 33 – the paragraph starting on this line is hard to parse, I suggest rewriting

Thanks, we rewrote as below in line 34:

"The long-term trends  the global carbon cycle are superimposed  with substantial year-to-year variations (Piao et al., 2020). These variations mostly  originate from natural processes,  including internal climate variability  –fluctuations across  a continuum of time scales  as well as from  natural external forcings such as volcanic eruptions and solar radiation (Deser et al., 2012b; Canadell et al., 2021; Eyring et al., 2021; Mercado et al., 2009; Zhang et al., 2021). Internal climate variability is often  regarded as irreducible noise  within the signal of long-term forced climatic trends. It  arises from internal atmospheric dynamics and  from atmosphere–ocean interactions (Deser et al., 2012a, 2020; Lehner et al., 2017; Bonan et al., 2021).  Such variability manifests both as short-term weather events and as low-frequency climate patterns, such as the El Niño–Southern Oscillation (ENSO), which  strongly influence global land carbon sink variations through associated  changes in temperature and precipitation."

Line 47 – can differ substantially from what? Please be specific

We change line 48 to: "First, internal climate variability can differ substantially in various simulations under the same external forcings, "

Line 50 – a good example of this is found in this paper:
https://egusphere.copernicus.org/preprints/2024/egusphere-2024-3684/

We added this citation in line 52: "…range of physically plausible internal climate variability. Moreover, internal climate variability is sensitive to the choice and length of the study period (Kumar et al., 2016; Doblas-Reyes et al., 2021; Maher et al., 2024)."

Line 53 – I would end this sentence with 'orividing a challenge because …'

In line 54:

" This makes it challenging to capture the full dynamics of internal climate variability, due to the limited length observation records."

Line 65 – this section is missing two key references:
https://agupubs.onlinelibrary.wiley.com/doi/10.1029/2019GB006453,
https://agupubs.onlinelibrary.wiley.com/doi/10.1029/2011gl050087

Thanks, we added accordingly in line 68: "Based on such large ensembles of ESM simulations, the "time of emergence (ToE)" can be determined as the time required for an external perturbed signal (anthropogenic-caused climate change) to become larger than the amplitude of natural variations (Hawkins and Sutton, 2012; Lehner et al., 2017; Schlunegger et al., 2020; Bonan et al., 2021)."

Line 73 – odd use of the word 'need'

We removed "need" in line 75: "driven by anthropogenic perturbations  to be detected"

**Methods**

Line 81 – in this section please add the number of ensemble members from each model

Thanks, we added below in line 85:

"The models selected include the CESM2-LE with 90 simulations (Danabasoglu et al., 2020; Rodgers et al., 2021) and four models in CMIP6 (Eyring et al., 2016; Brunner et al., 2020): ACCESS-ESM1-5 with 38 simulations (Ziehn et al., 2020), CanESM5 with 40 simulations (Swart et al., 2019), IPSL-CM6A-LR with 33 simulations (Boucher et al., 2020), and MPI-ESM1-2-LR with 41 simulations (Mauritsen et al., 2019)."

Line 82 – CESM2 is part of CMIP6

Here we use CESM2-LE dataset downloaded from NCAR directly: "https://rda.ucar.edu/gsearch/dataset-search/?q=CESM2-LE+", this dataset is a bit different from pre-processed CMIP6 by Brunner 2020, they only included 12 simulations in CMIP6.

Line 84 – this is the correct citation for MPI-ESM1-2-LR large ensemble: https://agupubs.onlinelibrary.wiley.com/doi/full/10.1029/2023MS003790

Thanks, but the simulations from MPI (MPI-ESM1-2-LR) and the other three CMIP6 models used in this study were pretreated by Brunner et al., 2020 (https://zenodo.org/records/3734128). Which means they are published before 2020. The citation you suggested is for the newest MPI (MPI-GE CMIP6) model and is published in 2023.

Line 125 – this is for CESM2 what about the other models?

We added other models in line 121: "Note that CESM2-LE only includes one future scenario (SSP3-7.0), and other models included all four future scenarios."

Line 133 – what about the SSPs

I explained SSPs in line 175, we have moved to line 136: "In the future scenarios, we calculate the ToE for NBP, GPP, and TER, with the signal period in 2020–2070 and the noise period in 2020–2070 (with the ensemble mean removed)."

Line 134 – how do you calculate N? do you pool the data? Do you take the STD of each member and average? Please provide this detail

We added in line 132: "For the calculation of N, we first gather a data pool including all ensemble residuals from simulations in the selected period, then mix the data from all years in the selected period together and calculate the standard deviation."

Line 137 – please add what 2N/S means in terms of significance of the emergence

We already explained in line 137: "ToE (Eq. (3)) represents the time needed for the anthropogenic perturbed signal to become larger than the amplitude of the noise (Bonan et al., 2021)."

"Line 147 – would 500mb geopotential height be a better proxy for circulation than SLP?

Yes, but sea level pressure (SLP) is physically more close to land ecosystems, and can reflect the direct influence of internal variability to land ecosystems.

Line 168 – you interchange 'ToE' and 'time to detect' this is confusing please stick to one terminology

We changed in line 177: "We analyse the  ToE of anthropogenic perturbed signal in NBP, GPP"

**Results:**

Line 213 – similar patterns to what? Each other?

Thanks, we changed to in line 210: "... both show similar patterns in detection time…"

Line 230 – grammar please fix 'takes shorter'

Thanks, we changed to in line 261: "We found  global scale takes shorter time to detect long-term trends induced by anthropogenic effects than at smaller scales,

Line 232 – please link with 'such as'

Thanks, we changed to in line 264: "  A few regions, however, show shorter ToEs than the global scale. For example, in Russia, CanESM5 and IPSL-CM6A-LR simulate relatively small noise and stronger signal trends, leading to shorter ToEs."

Line 233/234/236 – please reference the figures that you are discussing in this section

Thanks, we added.

Line 233 – remove 'is apparently'

We removed in line 264: "  A few regions, however, show shorter ToEs than the global scale. For example, in Russia, CanESM5 and IPSL-CM6A-LR simulate relatively small noise and stronger signal trends, leading to shorter ToEs."

Line 240 – is this statement true – I do not make the same conclusion from the Figure

You mean in Section 3.3  the first paragraph line 275: "We examine the time series of NBP under various future scenarios from 2016 to 2100 (Fig. 4). While NBP trends  show large deviations across models."

**Conclusions:**

Line 293 – you should be able to quantify if this is due to larger noise or not from your results not just speculate.

Thanks for the reviewer's concrete suggestion. Here we quantified the contribution of noise and signal to the regional increased ToE, in NBP, GPP, and TER. Please check our answers above in question 7.

Line 315 – can be detected how? There are no observations in the paper

Thanks, we have added the observed results as Figure 7 (as is shown above)

Additional changes:

1. Line 7-8: Our results show that, firstly, the anthropogenic signal in the global net land carbon sink emerges from 26 to 66 years in the period 1960–2009

2. Line 10: "Furthermore, we find that long-term trends of net land carbon sink on most regional scales take at least 20 years more to emerge, due to large contributions from internal climate variability and detected weak signals at smaller scales."

3. Line 65-66: "The externally perturbed signal (dominated by anthropogenic signal) emerges as the ensemble mean, that is, a deterministic signal."

4. Line 68-69: "the "time of emergence (ToE)" can be determined as the time required for an external perturbed signal (mostly anthropogenic-caused climate change) to become…"

5. Line 75-78: "Here, we evaluate how long it takes for long-term trends in the global land carbon sink—primarily driven by anthropogenic perturbations—to be detected  at different spatial scales.  To achieve this, we estimate the ToE in ESM simulations under historical and future scenarios. "

6. All "circulation induced" to "circulation-induced"

7. Line 155-156: "In our model, the sea level pressure (SLP) field is used as a predictor of NBP variability and as a proxy of  circulation-induced variability."

8. Line 156-157: "As a regularized linear regression method, ridge regression allows for including full spatiotemporal dynamics of circulation variations while overcoming multicollinearity and overfitting, which typically arise  from a large number of predictors and relatively short study period."

9. Line 160: "Select pixel based time series of global SLP, to be used later for predicting  global carbon cycle variability."

10. Line 162-163: " 2) Select the time series representing global land carbon  variability; here, this corresponds to the global annual NBP with the ensemble mean removed. "

11. We remove the definition of SSP, since the detailed discussion can be found in IPCC report in line 109.

    "The future scenario simulations are modeled under different Shared Socioeconomic Pathways (SSPs) for the period 2015-2100, based on varying levels of human-emitted CO2 and mitigation efforts (Chen et al., 2021; Lee et al., 2021; O'Neill et al., 2016).

12. Other values are slightly changed due to the changes of noise period from 2020-2050 to 2020-2070, in future scenarios. Also, the ToE values are cut until 150 years.

[revised manuscript text omitted]

---

## Author Comment (AC2)

**Reply to reviewers for the manuscript egusphere-2025-1924:"Constraining the time of emergence of anthropogenic signal in the global land carbon sink"**

**Na Li, Sebastian Sippel , Nora Linscheid , Miguel D. Mahecha , Markus Reichstein , and Ana Bastos**

We thank the reviewers for the productive and constructive comments. Here we reply to the comments in detail, and we suggest how to adjust the manuscript according to the comments. In addition, based on the comments, we made some extra changes in the manuscript for easier reading and understanding (at the last of each reply).

Below we address the changes according to the comments **(the line numbers in replies are according to the new manuscript that shows changes**). The blue colored lines are my replies, and the red words are locations changed.

**Changes according to the reviewer #2:**

**Specific comments:**

 General Comments

The manuscript examines the Time of Emergence (ToE) in both historical and future simulations derived from Earth System Models (ESMs), analyzing their spatial patterns and applying a dynamic adjustment approach to reduce the influence of circulation variability and thereby shorten detection time. The study addresses an important topic with potential implications for early climate change detection. However, the manuscript would benefit from a clearer explanation of the methods section and the addition of a dedicated discussion section. Several figures could also be improved to enhance clarity and interpretability. So I suggest a minor revision before the publication.

We thank the reviewer for the positive and constructive evaluation.

Specific comments

Title:

The use of the word "constraining" in the title may be somewhat misleading, as the manuscript primarily focuses on examining the Time of Emergence (ToE) and detection approaches, rather than directly constraining ToE. Consider revising the title to better reflect the core content and objectives of the study.

We thank the reviewer for pointing that out, we changed the title to: " Understanding the time of emergence of the anthropogenic signal in the global land carbon sink"

Methods:

Line 140: The meaning of "dynamic" in this context could be clarified. Is it related to dynamically adjusting the time window of the time series? Additionally, the connection between ridge regression and dynamic adjustment is unclear—does ridge regression serve as a method of dynamic adjustment here?

The dynamical means…, ridge regression is one methos of dynamic adjustment. We changed in line 154: "Here, we employ ridge regression, a  dynamic adjustment technique, to estimate…"

Line 235: This paragraph appears to overlap in content with the one beginning at line 219. Consider merging or clarifying to avoid redundancy.

Thanks for pointing this out, we remove the paragraph in line 273:

> "~~In the historical simulations, the land carbon sink (NBP) shows large year‑to‑year variability, delaying the detection of anthropogenic signals. In contrast, GPP and TER are primarily driven by anthropogenic perturbations, with relatively lower natural variability. The compensating trends of TER and GPP delay NBP detection, explaining why GPP and TER detect the signal in around 10 years, while NBP takes around 26 to 66 years. Next, we explore how the different future climate scenarios impact ToE.~~"

Section 2.4: The calculation of signal (S) and noise (N) would be clearer if accompanied by explicit equations and provided why linear regression is suitable for this purpose. Is S calculated as the slope of the regression of annual mean NBP versus year? And is N the standard deviation across all years (e.g., 1930–1959), or computed year-by-year? The definition of the time window in this section also seems inconsistent with the earlier discussion in Section 2.1.3. Further clarification would be helpful.

We thank the reviewer for highlighting this point, here we clarify the S and N definition in section 2.3.1 Line 130-136:

> "The signal (S) refers to the anthropogenic perturbation driven response, which is  the linear  regression slope of the ensemble mean of the simulations for each model (Bonan et al., 2021). For the calculation of N, we first gather a data pool including all ensemble residuals from simulations in the selected period, then mix the data from all years in the selected period together and calculate the standard deviation. In the historical simulations, the noise (N) corresponds to the standard deviation of the ensemble before the  1960s (here is 1930—1959), a period less affected by human activities compared to more recent ones, and used as the baseline for natural variability (Bonan et al., 2021)."

We also added one paragraph explaining why linear regression is suitable for this purpose. In line 140:

> "Here we use a linear regression slope rather than a nonlinear approach to represent the signal trend, the reasons are: 1) Capturing the dominant forced signal. The ensemble mean of NBP, GPP and TER reflects the forced ecosystem response, including anthropogenic forcing, short-period natural forcings (e.g., volcanic eruptions), and decadal internal variability (Deser et al., 2012b; Canadell et al., 2021; Eyring et al., 2021; Mercado et al., 2009; Zhang et al., 2021). The linear trend captures the most consistent anthropogenic influence, whereas nonlinear methods risk overfitting and misattributing natural or internal variability to anthropogenic signals, especially at regional scales where variability is larger (see Figure A.1 and Figure 3). "

Line 178: Are the reported values global means? It would be useful to clarify whether the reduction in detection time applies across all pixels in the study area or only a subset.

Additionally, do the same regions show a reduction after adjustment compared to before?

Thanks, all the reductions reported here are based on global scale, as we consider the test for regional and pixels beyond the scope of this study.  In line 184, we added:

> "4) Noise reduction through dynamical adjustment: Given the large year–to–year variability in NBP, we use ridge regression to remove the circulation induced variability in global NBP. To assess the effectiveness of ToE reduction on a global scale through dynamical adjustment…."

Line 182: Please confirm whether $V_0$ refers to the original ToE value or some other values. If it is the ToE value, the sentence could be rephrased for improved clarity:

"VR is the residual after the circulation-induced variability estimated by the ridge regression model is removed."

→ "VR is the ToE estimated from the residual time series after removing circulation-induced variability using the ridge regression model."

We thank the reviewer for pointing this out. We used this equation to calculate ToE reduction, but also for N reduction (as is shown in Table A.1-A.4). Here we clarify below in line 192:

> "Note that $V_O$ represents the original value (ToE or N) and $V_R$ is the (ToE or N) estimated from the original time series (NBP or GPP)  after removing the circulation-induced variability estimated by using the ridge regression model ."

Results:

Section 3.3: It might be more effective to show the future ToE results (currently in supplementary) as the main figure for this section, given that Section 3.3 primarily discusses ToE. Including such a figure could better support the narrative and conclusions.

Thanks. We added Figure A.9 (Heat map of ToE, noise, and signal of NBP under future scenarios.) up as new Figure 5. Note that we changed the noise period from 2020-2050 to 2020-2070, so the values in new Figure 5 are slightly different.

[Figure]

Figure 5. Heat map of ToE, noise, and signal of NBP under future scenarios.

Figure Clarifications

Figure 3: It would be helpful to include a legend or figure caption clarification for the lines shown. Indicating the meaning of the lines, units used for ToE, and spatial resolution would enhance the figure's interpretability.

Thanks for pointing out, we improved Figure 3 as below and also clarified details in the caption.

[Figure]

Figure 3. Spatial effect in NBP historical simulations across five ESM large ensembles. The distribution of ToE (years after 1960)  is shown for varying spatial resolutions. The line within each box indicates the median. Note that all  signals are in absolute values, so the calculated ToE are all positive.

Figure 5: The legend and color scheme are somewhat difficult to interpret. Consider using a more intuitive design—e.g., solid boxes for the original time series and hatched or patterned boxes for the residuals.

Thanks for suggesting this, we improved the Figure 5 (new Figure 6). Note that we changed the noise period from 2020-2050 to 2020-2070, and also only included ToEs with value less than 150 years. The similar plot for GPP also changed accordingly.

[Figure]

Figure 6. ToE of NBP from historical simulations to future scenarios. Note that ToE in historical simulations is calculated with signal period of 1960–2009 relative to the noise period of 1930–1959, and ToE in future scenarios is calculated with signal period of 2020–2070 relative to the noise period of 2020–2050, details please check Sect. 2.4. The  solid boxes represent the ToE of NBP, while the  hatched boxes represent the ToE of the NBP residual with the circulation induced variability removed. In cases where both boxes are missing, the respective  signal is not available (no significance of linear trend slope), or the ToEs are longer than 150 years.

Grammar/typo errors

Line 142: "circulation" → "Circulation"

Thanks, in line 150 we changed to: "Circulation induced variability"

Lines 152–157: Please revise for grammar.

Thanks, we have corrected in line 159 as:

"The key steps include (Sippel et al., 2019; Li et al., 2022): 1) Select pixel-based time series of global SLP to predict global carbon cycle variability. Then calculate the mean seasonal SLP.  Because DJF (December –February) SLP  provides the highest predictability of annual NBP ( Li et al. (2022) see Li et al., 2022 for details),  we use DJF SLP in this study. 2) Select the time series  representing global carbon cycle variability ; here, this corresponds to the global annual NBP with the ensemble mean removed. 3) Training and testing Split the dataset into training and testing groups; here,  the first half  of the dataset  is used for training group and the second half  for testing ."

Lines 168–169 and 174: These sentences contain grammatical issues and should be revised for clarity.

Line 176-177: "We perform four statistical analyses: 1) ToE in land carbon fluxes from historical simulations:. We analyze the ToE  of the anthropogenic perturbed signal in NBP, GPP, and TER in the historical simulations."

Line 180-182: "2) Spatial effects on ToE:. We examine how the ToE varies globally and across the 10 RECCAP-2 regions.  In addition, we evaluate the  influence of spatial resolution on ToE. We calculate  pixel-based ToE values at  multiple spatial scales (ranging from 2.5° × 2.5° to 60° × 60°) and compare  these with the global scale."

Line 102: "10distinct" → "10 distinct"

Thanks, we changed accordingly in line 106.

Additional changes.

1.  Line 7-8: Our results show that, firstly, the anthropogenic signal in the global net land carbon sink emerges from 26 to 66 years in the period 1960–2009

2.  Line 10: "Furthermore, we find that long-term trends of net land carbon sink on most regional scales take at least 20 years more to emerge, due to large contributions from internal climate variability and detected weak signals at smaller scales."

3.  Line 65-66: "The externally perturbed signal (dominated by anthropogenic signal) emerges as the ensemble mean, that is, a deterministic signal."

4.  Line 68-69: "the "time of emergence (ToE)" can be determined as the time required for an external perturbed signal (mostly anthropogenic-caused climate change) to become…"

5.  Line 75-78: "Here, we evaluate how long it takes for long-term trends in the global land carbon sink—primarily driven by anthropogenic perturbations—to be detected  at different spatial scales.  To achieve this, we estimate the ToE in ESM simulations under historical and future scenarios. "

6.  All "circulation induced" to "circulation-induced"

7.  Line 155-156: "In our model, the sea level pressure (SLP) field is used as a predictor of NBP variability and as a proxy of  circulation-induced variability."

8.  Line 156-157: "As a regularized linear regression method, ridge regression allows for including full spatiotemporal dynamics of circulation variations while overcoming multicollinearity and overfitting, which typically arise  from a large number of predictors and relatively short study period."

9.  Line 160: "Select pixel based time series of global SLP, to be used later for predicting  global carbon cycle variability."

10. Line 162-163: " 2) Select the time series representing global land carbon  variability; here, this corresponds to the global annual NBP with the ensemble mean removed. "

11. We remove the definition of SSP, since the detailed discussion can be found in IPCC report in line 109.

    "The future scenario simulations are modeled under different Shared Socioeconomic Pathways (SSPs) for the period 2015-2100, based on varying levels of human-emitted CO2 and mitigation efforts (Chen et al., 2021; Lee et al., 2021; O'Neill et al., 2016).

12. Other values are slightly changed due to the changes of noise period from 2020-2050 to 2020-2070, in future scenarios. Also, the ToE values are cut until 150 years.